# Federated Data and Feature Selection by Generalized CUR Decomposition

Ying-Peng Tang [1]    Zhuang Qi [2]    Xiaoli Tang [1]    Wei Zhuo [1]    Sheng-Jun Huang [3]    Han Yu [1]

## Abstract

With the advance of federated learning (FL) in privacy-sensitive domains, the need for efficient and robust training becomes increasingly urgent. Communication bottlenecks, heterogeneous client distributions, and fairness requirements make it essential to select the "right" data and features for model training. Yet existing FL research often addresses feature selection and data selection separately, ignoring their interplay in real-world high-dimensional and noisy datasets, leading to suboptimal performance. In this paper, we propose a unified framework for data and feature selection by formulating the problem as a generalized CUR decomposition problem. We introduce FedGCUR, a practical framework that integrates a federated column-pivoted QR (FedCPQR) decomposition routine with per-silo row selection. We prove that FedCPQR produces exactly the same decomposition results as centralized CPQR and establish an upper bound of the reconstruction error of FedGCUR. Experiments on tabular tasks and vision benchmarks show strong downstream accuracy and absolute CUR reconstruction quality compared with representative data and feature selection baselines.

## 1. Introduction

Federated learning (FL) (Zhang et al., 2021) has emerged as a key paradigm for collaborative model development across data silos and edge devices, enabling training without centralizing sensitive records (Kairouz et al., 2021; Li et al., 2026). By joint local learning and global model aggregation, FL effectively exploits the collective knowledge of distributed clients in a privacy-preserving manner (Li

et al., 2020; Meng et al., 2025; Qi et al., 2025). However, several practical obstacles constrain the efficacy of current FL frameworks. Specifically, communication overhead often acts as the primary bottleneck in real-world applications (Niknam et al., 2020; Guo et al., 2025). Furthermore, there is a frequent necessity to "sketch" local or aggregated datasets, for instance, to facilitate the sharing of low-rank representations. These challenges are significantly exacerbated in settings characterized by statistical heterogeneity (i.e., non-IID data) across silos (Qi et al., 2023; Ye et al., 2023). Consequently, these constraints necessitate the development of effective selection of the most informative data and features for federated learning.

Recently, numerous studies in federated learning have investigated data and feature selection (Fu et al., 2023; Hu et al., 2023; Wang et al., 2024). The main idea is to select a compact set of globally shared features to reduce client-side computation and uplink communication costs. Under participation or communication constraints, selecting more informative samples can further improve model prediction and distillation performance. Existing studies typically consider feature selection (Fu et al., 2023; Zhang et al., 2023; Hermo et al., 2024) and data selection (Liu et al., 2022; Li et al., 2021; Rizk et al., 2022) as isolated subproblems rather than as interdependent decisions within a unified framework. They often directly transfer centralized techniques to cross-silo scenarios, without fully accounting for the heterogeneity inherent in FL. However, feature and data selection are coupled in many applications (Garcia-Pedrajas et al., 2021; Dornaika, 2021; Fan et al., 2022), e.g., in a high-dimensional datasets with many noisy and redundant data points. In such settings, if one axis is optimized in isolation, the result can be suboptimal or even contradictory.

One of the highly related techniques for data and feature selection is the matrix decomposition (Golub & Van Loan, 2013), which is fundamental to machine learning and data analysis (Caiafa et al., 2020; Sidiropoulos et al., 2017; Li et al., 2019). They are useful in dimension reduction, principal analysis, coreset discovery, etc. Unfortunately, most methods assume centralized access to the full matrix and thus do not directly apply in FL due to privacy and bandwidth constraints. Recent work has made preliminary progress on federated matrix decomposition (Chai et al., 2022; Hartebrodt & Röttger, 2023) by combining secure ag-

---

[1]College of Computing and Data Science, Nanyang Technological University, Singapore [2]School of Software, Shandong University, Jinan, China [3]College of Computer Science and Technology, Nanjing University of Aeronautics and Astronautics, Nanjing, China. Correspondence to: Han Yu <han.yu@ntu.edu.sg>.

*Proceedings of the 43rd International Conference on Machine Learning*, Seoul, South Korea. PMLR 306, 2026. Copyright 2026 by the author(s).

gregation and data masking to avoid exposing raw data, but more expressive federated matrix decompositions remain underexplored.

Among decomposition-based selection methods, CUR (Mahoney & Drineas, 2009; Li et al., 2019) is particularly appealing. Given a matrix $A$, CUR seeks factors $A \approx CMT$, where $C$ collects a subset of columns (features), $T$ collects a subset of rows (instances), and $M$ is a small linking matrix. In FL, however, CUR must satisfy federated constraints that do not arise centrally: (i) select a shared set of features so that a joint model can be trained and aggregated across silos; (ii) reveal only privacy-preserving aggregated statistics. These requirements make the federated CUR problem challenging, because rank-revealing selection typically needs global coordination; and heterogeneous silos may prefer different pivots; and naive protocols either leak information or incur prohibitive costs.

In this paper, we bridge this gap by formulating federated data and feature selection as a federated generalized CUR (FedGCUR) decomposition problem. To the best of our knowledge, this is the first work that considers both data and feature selection in FL setting. First, we develop a federated column-pivoted QR decomposition routine (FedCPQR) that computes a global column pivot order using only secure sums of squared norms and inner products, with no raw data sharing. Second, we build FedGCUR on top of this routine: it uses the global pivots to select a common set of columns and then performs per-silo row selection locally, yielding CUR factors that approximate each silo's data. Theoretical analyses are conducted to show that the proposed FedCPQR outputs exactly the same results as the centralized modified Gram-Schmidt based CPQR, and to establish reconstruction guarantees for FedGCUR, including a per-silo bound that explicitly reflects how well the global feature set serves each silo. The experimental results on six OpenML tabular datasets and Fashion-MNIST, CIFAR-10, CIFAR-100, and Tiny ImageNet show that FedGCUR achieves strong reconstruction quality and competitive downstream accuracy compared with data and feature selection baselines.

## 2. Related Works

**Federated data selection and feature selection** Recent work explores privacy-preserving data and feature selection in federated settings (Fu et al., 2023; Zhang et al., 2023; Li et al., 2021; Rizk et al., 2022). For example, Hu et al. (2023) propose a heuristic federated feature selection scheme where each client performs local subset selection and shares only the selected feature indices and their local accuracy, after re-evaluating others' subsets, and the aggregator selects a global subset via an aggregation rule. Wang et al. (2024) propose FAST, a federated learning framework that jointly adapts class-wise data sampling and the number of local iterations to counter Non-IID data and device heterogeneity, with an online MAB-based controller and a proved convergence bound. Hermo et al. (2024) propose Fed-mRMR, a lossless federated adaptation of the classic mRMR filter that replaces raw-data sharing with an occurrences matrix built from local bitmaps and merged across clients, preserving the exact mRMR ranking under non-IID, cross-silo FL. Most existing methods focus on either data selection or feature selection in isolation, which can be suboptimal when both are required, as they ignore the coupling between instances and features.

**Federated matrix decomposition** Matrix decomposition in federated learning receives less attention. Strong privacy constraints preclude sharing matrix statistics, making standard decompositions difficult to adapt. Hartebrodt & Röttger (2023) studied federated QR (FedQR) decomposition. A modified Gram-Schmidt based QR decomposition method is proposed for federated learning settings. While effective for basic factorization, FedQR is non-pivoted, meaning the resulting decomposition depends entirely on the input ordering of features. Consequently, it cannot be used for feature selection, sparse approximation, or rank determination. Chai et al. (2022) investigate FedSVD for vertical FL. A trusted authority generates a unitary random mask, clients upload masked data, and the server computes an SVD on the masked matrix under secure aggregation. Each client receives the right singular vectors corresponding to its features, while the left singular vectors and singular values are shared globally. Tensor decomposition in FL (Gao et al., 2021; Ouyang et al., 2024) has also been explored, but it lies beyond the scope of this paper.

## 3. Methodology

### 3.1. Preliminaries

**Notations** In this paper, matrices or sets are denoted by capital letters, vectors by bold lowercase letters, and scalars by lowercase letters. We use superscript $\dagger$ to represent the pseudo-inverse of a matrix, and $\sigma_j$ represents the $j$-th largest singular value. An $m \times m$ identity matrix is denoted by $I_m$, and we omit the subscript when the context is clear. Denote by $\|\cdot\|_2$ the spectral norm for matrix and $\ell_2$ norm for vector, and $\|\cdot\|_F$ the Frobenius norm. We define $[n] = \{1, 2, \ldots, n\}$.

This work focuses primarily on cross-silo horizontal FL scenarios. Specifically, given a data matrix $A \in \mathbb{R}^{n \times d}$, where each row corresponds to one sample and each site owns a subset of rows, we write $A = \left[ A_1^\top, \ldots, A_s^\top \right]^\top$, with $A_c \in \mathbb{R}^{n_c \times d}$, and $\sum_{c=1}^s n_c = n$. Here, $c$ represents the client index. We assume that each row of $A$ is randomly sampled from an unknown distribution, though the split function across clients is arbitrary. The target vector is

denoted as $\boldsymbol{b} \in \mathbb{R}^{n \times 1}$. Throughout, $k$ is the target column rank, $j$ represents the column index, and $r_c$ is the number of selected data points in silo $c$, with $r = \sum_{c=1}^{s} r_c$ and $r_c \leq n_c$.

**Secure Aggregation Protocol**   We follow (Hartebrodt & Röttger, 2023) to use the additive secure aggregation protocol introduced by (Cramer et al., 2015), which, for a set of scalar values $\{x_c\}_{c=1}^{s}$, securely returns the sum $\sum_{c=1}^{s} x_c$ to all participants without revealing the individual $x_c$ values. This is mainly achieved by introducing a large prime known to all participants and transmitting the remainder of the data to be aggregated.

**CUR and Generalized CUR Decomposition**   CUR decomposition (Mahoney & Drineas, 2009) seeks a subset of $k$ columns and $r$ rows from $A \in \mathbb{R}^{n \times d}$ such that $A \approx CMT = ADMS^\top A$, where $C \in \mathbb{R}^{n \times k}$ and $T \in \mathbb{R}^{r \times d}$ are sub-matrices of the columns and rows of $A$, respectively, and $M \in \mathbb{R}^{k \times r}$ is the middle factor. Here, $D \in \mathbb{R}^{d \times k}$ and $S \in \mathbb{R}^{n \times r}$ are index selection matrices, constructed from columns of the identity matrix $I$ with appropriate permutations.

Generalized CUR decomposition is usually defined on a pair of matrices (Gidisu & Hochstenbach, 2022; Cao et al., 2024). Here, we generalize its definition to $s$ matrices: $A_1, \ldots, A_s$. The formal definition is as follows:

**Definition 3.1** (Generalized CUR Decomposition for Multiple Matrices). Let $s \in \mathbb{N}, s > 1$, and $c \in [s]$, $\{A_c\}_{c=1}^{s}$ be a collection of matrices where $A_c \in \mathbb{R}^{n_c \times d}$. Given the number of sampled rows $r_c$ for $A_c$ (with $r_c \leq n_c$) and the number of sampled columns $k$ (with $k \leq d$), the generalized CUR decomposition of $\{A_c\}_{c=1}^{s}$ seeks approximations of the form

$$A_c \approx C_c M_c T_c = A_c D M_c S_c^\top A_c, \qquad (1)$$

where, $S_c \in \mathbb{R}^{n_c \times r_c}$ and $D \in \mathbb{R}^{d \times k}$ are index selection matrices. When $s = 2$, the decomposition problem coincides with Gidisu & Hochstenbach (2022).

It is important to note that all $\{A_c\}_{c=1}^{s}$ share the same index selection matrix $D$ in the generalized CUR decomposition.

## 3.2. Federated CPQR

Considering federated data and feature selection, it is necessary to perform feature selection globally without exposing the raw data held by each client. This task can be formulated as a federated column-pivoted QR decomposition problem. The primary difference between standard QR (Watkins, 1982) and column-pivoted QR (CPQR) (Quintana-Ortí et al., 1998) lies in the column selection strategy. Specifically, let $U = [U_1, \ldots, U_s]^\top$ be the working matrix. At iteration $i$,

CPQR selects the column with the largest squared norm from the remaining unprocessed columns as the next pivot: $p = \arg\max_{j \in \{i, \ldots, d\}} \|U(:, j)\|_2^2$, and swaps it into the $i$-th position. The $i$-th column is then normalized:

$$Q(:, i) = \frac{U(:, i)}{\|U(:, i)\|_2}, \qquad R_{ii} = \|U(:, i)\|_2. \quad (2)$$

All remaining columns $j > i$ are orthogonalized against $Q(:, i)$:

$$R_{ij} = Q(:, i)^\top U(:, j), \qquad (3)$$
$$U(:, j) \leftarrow U(:, j) - Q(:, i) R_{ij}. \qquad (4)$$

However, centralized CPQR decomposition algorithm can not be applied directly due to the privacy constraints. To the best of our knowledge, CPQR in the context of FL remains largely unexplored.

In FL, data is distributed across multiple parties, i.e., $A = \left[A_1^\top, \ldots, A_s^\top\right]^\top$, and FedCPQR seeks the first $k$ pivots of a partial decomposition $AP \approx QR$, where $P$ is a permutation matrix reordering columns of $A$; $Q = \left[Q_1^\top, \ldots, Q_s^\top\right]^\top$ has orthonormal columns, and $R$ is upper triangular up to the computed rank. Designing FedCPQR therefore requires addressing both distributed data partitioning and privacy preservation. To address these challenges, we propose to build on the modified Gram-Schmidt process to iteratively compute an orthogonal basis, while incorporating column pivoting in FL. The key observation is that both the pivoting and orthogonalization steps can be expressed as sums of local scalar quantities:

$$\|U(:, j)\|_2^2 = \sum_{c=1}^{s} \|U_c(:, j)\|_2^2, \qquad (5)$$

$$Q(:, i)^\top U(:, j) = \sum_{c=1}^{s} Q_c(:, i)^\top U_c(:, j). \qquad (6)$$

These sums can be computed using secure aggregation such that no site exposes its raw data. After the requested $k$ iterations, the partial factor $R$ and permutation $P$ are known globally, while each site retains $Q_c$ that corresponds to its local data. Additionally, the squared norms of the remaining columns are updated after each deflation step to maintain accurate pivot selection:

$$d_j = \sum_{c=1}^{s} \|U_c(:, j)\|_2^2, \quad d_j \leftarrow d_j - r_{ij}^2, \qquad (7)$$

$$r_{ij} = \sum_{c=1}^{s} Q_c(:, i)^\top U_c(:, j). \qquad (8)$$

The main procedures of FedCPQR are summarized in Alg. 1. The SECAGG($\cdot$) represents the additive secure aggregation operation, which returns the sum across sites to all participants in clear text.

*Table 1.* Key differences between FedQR and FedCPQR.

| | **FedQR** (Hartebrodt & Röttger, 2023) |
|---|---|
| Rows of $A$ | Kept local |
| Factor $Q_c$ | Kept local |
| Shared factor | $R$ |
| Also revealed | Global norms / inner products |
| Implied leakage | $A^\top A = R^\top R$ |
| Special-case risk | Raw data may leak if only 2 parties |
| | **FedCPQR** (Proposed in this paper) |
| Rows of $A$ | Kept local |
| Factor $Q_c$ | Kept local |
| Shared factor | $R, P$ |
| Also revealed | Global norms / inner products and pivot residual norms |
| Implied leakage | $A^\top A = P R^\top R P^\top$ |
| Special-case risk | Raw data may leak if only 2 parties |

### 3.2.1. PRIVACY CONCERN

While $\text{SECAGG}(\cdot)$ prevents the server from inspecting individual updates, the final FedCPQR output reveals the permutation $P$ and triangular factor $R$. If the factorization is run to completion, then, as in FedQR (Hartebrodt & Röttger, 2023), the released factors determine the global Gram matrix $G = A^\top A = P R^\top R P^\top$. In the truncated feature-selection setting, they reveal the corresponding pivoted rank-$k$ Gram information. Table 1 summarizes the difference between FedQR and FedCPQR.

To further enhance the privacy preserving of FedCPQR, a practical mitigation is to combine secure aggregation with differential privacy (DP) technique (Dwork & Roth, 2014). The intuition is simple: secure aggregation hides each client's summand, whereas DP limits what can be inferred from the aggregate itself. For every scalar released by FedCPQR, let $q_t(A)$ denote either a residual squared-norm sum or an inner-product sum at release step $t$. Before aggregation, each record-level scalar contribution is clipped: squared-norm contributions are clipped to $[0, B_{\text{norm}}]$ and inner-product contributions to $[-B_{\text{prod}}, B_{\text{prod}}]$. Under add/remove record adjacency, the corresponding scalar sensitivities satisfy $S_t \leq B_{\text{norm}}$ or $S_t \leq B_{\text{prod}}$; under replace-one adjacency, these bounds are multiplied by two. The Gaussian mechanism (Dwork & Roth, 2014) releases

$$\widetilde{q}_t(A) = q_t(A) + Z_t, \qquad Z_t \sim \mathcal{N}(0, \sigma_t^2), \qquad (9)$$

and gives $(\epsilon_t, \delta_t)$-DP for that scalar release whenever $0 < \epsilon_t \leq 1$ and

$$\sigma_t \geq \frac{S_t \sqrt{2 \log(1.25/\delta_t)}}{\epsilon_t}. \qquad (10)$$

Thus the noise scale is the standard Gaussian-mechanism calibration. If all $T = d + \sum_{i=1}^k (1 + d - i)$ norm and

---

**Algorithm 1** FedCPQR

**Input:** Each site $c \in [s]$ holds $A_c \in \mathbb{R}^{n_c \times d}$; target rank $k \leq d$.
1: **Initialize:** $P \leftarrow I_d$, $R \leftarrow 0_{k \times d}$. $U_c \leftarrow A_c$, for $c \in [s]$.
2: For each $j \in [d]$, $c \in [s]$: compute $d_{j,c} \leftarrow \|U_c(:,j)\|_2^2$; $d_j \leftarrow \text{SECAGG}(\{d_{j,c}\}_{c=1}^s)$.
3: **for** $i = 1$ **to** $k$ **do**
4: $\quad p \leftarrow \arg\max_{j \in \{i,\ldots,d\}} d_j$. Swap positions $i \leftrightarrow p$ in $P$, in each local $U_c$, in $d$, and in the first $i{-}1$ rows of $R$.
5: $\quad$ Each $c$ computes $n_{i,c} \leftarrow \|U_c(:,i)\|_2^2$; all get $n_i \leftarrow \text{SECAGG}(\{n_{i,c}\})$.
6: $\quad Q_c(:,i) \leftarrow U_c(:,i)/\sqrt{n_i}$; set $R_{ii} \leftarrow \sqrt{n_i}$.
7: $\quad$ **for each** $j \in \{i+1, \ldots, d\}$ **do**
8: $\quad\quad r_{ij,c} \leftarrow Q_c(:,i)^\top U_c(:,j)$; all get $r_{ij} \leftarrow \text{SECAGG}(\{r_{ij,c}\})$; $R_{ij} \leftarrow r_{ij}$.
9: $\quad\quad U_c(:,j) \leftarrow U_c(:,j) - Q_c(:,i)\, r_{ij}$.
10: $\quad\quad d_j \leftarrow d_j - r_{ij}^2$.
11: $\quad$ **end for**
12: **end for**
13: **Return to all sites:** $P, R$; **each site $c$ keeps:** $Q_c$.

---

inner-product scalar releases are privatized, basic sequential composition gives $(\sum_t \epsilon_t, \sum_t \delta_t)$-DP. A tighter common accounting follows the advanced composition theorem of Dwork, Rothblum, and Vadhan (Dwork & Roth, 2014; Dwork et al., 2010, Theorem 3.20): if each scalar release is $(\epsilon_0, \delta_0)$-DP, then for any $\delta' > 0$ the composed FedCPQR transcript is

$$\left( \sqrt{2T \log(1/\delta')}\, \epsilon_0 + T\epsilon_0 (e^{\epsilon_0} - 1), \ T\delta_0 + \delta' \right) \text{-DP}. \tag{11}$$

This privacy layer perturbs pivot scores and projections, so it trades exact QR equivalence for protection against inference from the released aggregates. Appendix D reports reconstruction-attack simulations under this DP mechanism.

### 3.2.2. ANALYSES OF FEDCPQR

**Communication Cost** At each iteration $i$, FedCPQR relies on secure aggregation of scalars: (i) the pivot column norm $n_i$; (ii) the projections $r_{ij}$ for all remaining columns $j > i$. Thus, iteration $i$ aggregates $1 + (d - i)$ scalars after the initial column-norm aggregation. Across $k$ pivot steps, this gives $\mathcal{O}(kd)$ scalar aggregations, and the full factorization case $k = d$ gives $\mathcal{O}(d^2)$. The cost is independent of the number of samples $n$, making FedCPQR suitable for horizontal federated learning with large local datasets.

**Computation Cost** Each site performs standard local operations on its data $A_c \in \mathbb{R}^{n_c \times d}$: (i) computing local column norms $\|U_c(:,j)\|_2^2$ for active positions $j$, (ii) computing local projections $Q_c(:,i)^\top U_c(:,j)$, (iii) performing local de-

flation $U_c(:, j) \leftarrow U_c(:, j) - Q_c(:, i)r_{ij}$. At iteration $i$, the local computation cost is $\mathcal{O}\big(n_c(d - i)\big)$ for projections and deflations, leading to a total per-site cost of $\mathcal{O}(n_c dk)$ for a rank-$k$ run and $\mathcal{O}(n_c d^2)$ for a full run. This matches the corresponding centralized CPQR arithmetic and scales linearly with local data size, making FedCPQR computationally practical in FL.

**Correctness** All pivot decisions and residual updates depend solely on sums of squared norms and inner products. These quantities decompose additively and can be securely aggregated, ensuring that FedCPQR reproduces the exact centralized CPQR on $A$. Formally, we state the following equivalence, the proof is deferred to appendix.

**Theorem 3.2.** *Assume exact arithmetic, exact secure aggregation of all required inner products and squared norms, deterministic tie-breaking for equal pivot norms, and nonzero pivot norms for the first $k$ iterations. Then the first $k$ pivot choices, triangular entries, local orthogonal blocks, and residual updates of FedCPQR are identical to those of centralized modified Gram-Schmidt based CPQR applied to $A = \big[A_1^\top, \ldots, A_s^\top\big]^\top$.*

## 3.3. Federated Generalized CUR Decomposition

Using the proposed FedCPQR as a sub-routine, we then propose the feature and data selection method based on FedGCUR algorithm. Specifically, FedGCUR uses one common column selection matrix across all parties: it first runs the FedCPQR (Alg. 1) to obtain the global permutation $P \in \mathbb{R}^{d \times d}$ and set

$$D = P(:, 1 : k) \in \mathbb{R}^{d \times k}, \qquad (12)$$

$$C = AD \in \mathbb{R}^{n \times k}, \quad C_c = A_c D \in \mathbb{R}^{n_c \times k}. \qquad (13)$$

Every silo then performs a local exact CPQR of $C_c^\top$ to select $r_c$ informative rows. $S_c \in \{0, 1\}^{n_c \times r_c}$ is a row-selection matrix that picks those row indices (so $S_c^\top C_c \in \mathbb{R}^{r_c \times k}$ contains the selected rows of $C_c$), and define the local row submatrix $T_c := S_c^\top A_c \in \mathbb{R}^{r_c \times d}$. Finally, each silo computes its middle factor $M_c := C_c^\dagger A_c T_c^\dagger \in \mathbb{R}^{k \times r_c}$. The blockwise reconstruction stacks the local CURs:

$$\widehat{A}_{\text{loc}} = \begin{bmatrix} C_1 M_1 T_1 \\ \vdots \\ C_s M_s T_s \end{bmatrix} = \text{blkrow}_{c=1}^s \big(C_c M_c T_c\big).$$

The main steps of the proposed FedGCUR are summarized at Alg. 2. Note that, calculating the middle factor $M_c$ is optional. In the scenarios that only considers the feature and data selection, one can skip this step to reduce the computational cost.

---

**Algorithm 2** Data and Feature Selection by FedGCUR.

**Input:** $A = [A_1^\top, \cdots, A_s^\top]^\top \in \mathbb{R}^{n \times d}$; target rank $k$; per-silo quota $r_c \geq 1, c \in [s]$.
1: Run FedCPQR (Alg. 1) on $A$ to obtain the $k$ pivot columns $D$ and set $C_c := A_c D \in \mathbb{R}^{n_c \times k}$.
2: Each silo $c$ runs exact CPQR locally on $C_c^\top$ and outputs $r_c$ pivots $S_c$, and set $T_c := S_c^\top A_c$.
3: (Optional) Compute $M_c \approx C_c^\dagger A_c T_c^\dagger$.
4: **Return:** global $D$; each silo $c$ keeps $T_c, M_c, C_c$.

---

### 3.3.1. ANALYSIS OF FEDGCUR

**Computation, Communication and Privacy:** Along with the cost of FedCPQR, each silo performs an exact CPQR locally on $C_c^\top \in \mathbb{R}^{k \times n_c}$, which costs $\mathcal{O}(n_c k^2)$ flops to select rows. These costs match the corresponding centralized arithmetic applied to the vertically stacked $A$, up to negligible coordination overhead. Regarding the communication cost, no communication is needed for the local row selection and local $M_c$ formation. Therefore, FedGCUR does not introduce extra communication overhead, neither incurring any additional privacy concerns.

**Reconstruction Error Analysis** We establish the reconstruction error bound for FedGCUR. Denote the local column projector by $P_{C,c} := C_c C_c^\dagger$ and the local row projector by $P_{R,c} := T_c^\dagger T_c$. For each silo,

$$\widehat{A}_c = C_c M_c T_c = P_{C,c} A_c P_{R,c}. \qquad (14)$$

Let $P_k := DD^\top$ be the orthogonal projector onto the $k$ selected coordinate axes, and define the local oblique projector in the feature-index space by

$$P_{X,c} := D(A_c D)^\dagger A_c = DC_c^\dagger A_c \in \mathbb{R}^{d \times d}. \qquad (15)$$

We write $\eta_c := \|I_d - P_{X,c}\|_2$ and

$$\rho_{c,\xi} := \|P_{C,c} A_c (I_d - P_{R,c})\|_\xi, \qquad \xi \in \{2, F\}. \qquad (16)$$

Here, $\eta_c$ measures how strongly the globally selected columns amplify the unselected-feature residual of silo $c$, while $\rho_{c,\xi}$ is the remaining local row-selection residual after projecting onto the selected-column subspace. We have the following analyses on the reconstruction error of FedGCUR:

**Theorem 3.3.** *Let $A \in \mathbb{R}^{n \times d}$ be partitioned by rows into $s$ parties, $A = \big[A_1^\top, \ldots, A_s^\top\big]^\top$, with $A_c \in \mathbb{R}^{n_c \times d}$. Let $D \in \mathbb{R}^{d \times k}$ be the column selection matrix returned by FedCPQR, let $C_c = A_c D$, let $T_c = S_c^\top A_c$, and let $\widehat{A}_c = C_c C_c^\dagger A_c T_c^\dagger T_c$ be the FedGCUR reconstruction for silo c. Then, for each silo c and each $\xi \in \{2, F\}$,*

$$\|A_c - \widehat{A}_c\|_\xi \leq \eta_c \|A_c(I_d - P_k)\|_\xi + \rho_{c,\xi}. \qquad (17)$$

Consequently, for the stacked reconstruction $\widehat{A}_{\text{loc}} = \left[\widehat{A}_1^\top, \ldots, \widehat{A}_s^\top\right]^\top$,

$$\|A - \widehat{A}_{\text{loc}}\|_\xi \leq \left(\sum_{c=1}^s \left[\eta_c \|A_c(I_d - P_k)\|_\xi + \rho_{c,\xi}\right]^2\right)^{1/2}. \tag{18}$$

Here, $P_k = DD^\top$ is the orthogonal projector onto the selected coordinate axes, $P_{X,c} = D(A_c D)^\dagger A_c$, $\eta_c = \|I_d - P_{X,c}\|_2$, and $\rho_{c,\xi} = \|P_{C,c} A_c(I_d - P_{R,c})\|_\xi$.

*Proof sketch.* For each silo, decompose the local error as $A_c - \widehat{A}_c = (I - P_{C,c})A_c + P_{C,c}A_c(I - P_{R,c})$. The second term is exactly $\rho_{c,\xi}$. For the first term, use $(I - P_{C,c})C_c = 0$, which implies $(I - P_{C,c})A_c = (I - P_{C,c})A_c(I - P_k)$. Since $I - P_{C,c}$ is an orthogonal projector, this gives a contractive column-residual bound by $\|A_c(I - P_k)\|_\xi$. Moreover, $A_c P_{X,c} = P_{C,c} A_c$ and $P_{X,c}$ is idempotent; hence $\eta_c = \|I_d - P_{X,c}\|_2$ is either $0$ in the exact-interpolation case or at least $1$. Stacking the silo errors gives Eq. (18). The full proof is given in Appendix C. $\square$

**Remark 1.** Theorem 3.3 shows that FedGCUR's error is controlled by a silo-specific column residual and a local row residual. The term $\|A_c(I_d - P_k)\|$ measures how much of silo $c$'s data lies in globally unselected features, and $\eta_c$ measures the amplification caused by applying the global pivots to that silo. This makes cross-silo heterogeneity explicit: a silo whose important features are not well aligned with the global feature set will have a larger bound.

# 4. Experiment

## 4.1. Empirical Settings

We conduct extensive experiments on a diverse set of datasets to evaluate the effectiveness and robustness of our proposed methods, FedCPQR and FedGCUR, across varying domains and data characteristics. Specifically, we use six datasets from the OpenML (Vanschoren et al., 2014) repository: mfeat-pixel, gina_prior2, devnagari-script, USPS, guillermo, and isolet, which differ in dimensionality and sample size. Table 2 summaries the main information.

To assess the correctness of FedCPQR, we adopt three performance metrics: pivot exact match, maximum principal angle, and projection distance, and compare our method against the QR decomposition with column pivoting implemented in SciPy (Virtanen et al., 2020) (i.e., `scipy.linalg.qr`). Specifically, let $AP_{fed} = Q_{fed}R_{fed}$ and $AP_{sci} = Q_{sci}R_{sci}$ denote the decompositions obtained by FedCPQR and SciPy, respectively. The metrics are defined as follows:

• **Pivot exact match** checks whether the set of selected

*Table 2.* OpenML Datasets Summary.

| Dataset (ID) | # Train | # Test | # Fea. | # Lab. |
|---|---|---|---|---|
| mfeat-pixel (1022) | 1600 | 400 | 240 | 2 |
| gina_prior2 (1041) | 2774 | 694 | 784 | 10 |
| devnagari (40923) | 73600 | 18400 | 1024 | 46 |
| USPS (41082) | 7438 | 1860 | 256 | 10 |
| guillermo (41159) | 16000 | 4000 | 4296 | 2 |
| isolet (43985) | 6237 | 1560 | 613 | 26 |

columns by FedCPQR exactly matches the first $k$ pivot columns chosen by SciPy (i.e., comparing $P_{fed}$ and $P_{sci}$).

• **Maximum principal angle** measures the largest angular deviation between the subspaces spanned by $Q_{fed}$ and $Q_{sci}$, characterizing worst-case misalignment.

• **Projection distance** is computed as $\|Q_{fed}Q_{fed}^\top - Q_{sci}Q_{sci}^\top\|_F$, quantifying the aggregate deviation of $Q_{fed}$ from the reference subspace.

To evaluate the data and feature selection capability of FedGCUR, we use the selected subsets to train a global model and compare its accuracy against combinations of existing approaches: **Data selection:** Coreset (Sener & Savarese, 2018), Leverage Score (Larsen & Kolda, 2022), and Random. **Feature selection:** Maximum variance selection and Random. Note that data selection is performed locally at each party, with each selecting half of its local samples. For Leverage Score sampling, the selection scores are computed using the globally estimated $\left[Q_1^\top, \ldots, Q_s^\top\right]^\top$ from FedCPQR, which can be viewed as a degenerate variant of our method in the federated setting.

We empirically set the federated parties as 10. A three-layer neural network is employed as target model. FedAvg (McMahan et al., 2017) is used to aggregate the global model with a communication round of 10. The target decomposition column rank is selected from $k \in \{10, 50, 100\}$. We study two data split settings in the learning task experiment: uniform partitioning (i.i.d. split), and Dirichlet distribution-based partitioning (non-i.i.d. split) (Yurochkin et al., 2019) with concentration parameter $\alpha = 0.5$, where smaller $\alpha$ values correspond to higher data heterogeneity.

## 4.2. Results

**Correctness of FedCPQR** We report the correctness results of FedCPQR in Table 3. The dataset is split i.i.d. for FL in this experiment. For all datasets and target ranks, the pivoted columns selected by FedCPQR exactly match those obtained by SciPy. The differences in maximum principal angle and projection distance are negligible, on the order of $10^{-14}$, and can be attributed to numerical precision loss. These findings confirm that FedCPQR achieves CPQR de-

*Table 3.* The decomposition results precision comparison between FedCPQR and SciPy.qr function.

| Dataset | Pivot Exactly Match | | | Max Principal Angle | | | Projection Dist. (Frobenius) | | |
| ID | k=10 | k=50 | k=100 | k=10 | k=50 | k=100 | k=10 | k=50 | k=100 |
| --- | --- | --- | --- | --- | --- | --- | --- | --- | --- |
| 1022 | True | True | True | 1.19e-15 | 5.97e-15 | 1.15e-14 | 2.27e-15 | 1.20e-14 | 2.52e-14 |
| 1041 | True | True | True | 1.02e-14 | 1.02e-14 | 1.02e-14 | 1.87e-15 | 1.35e-14 | 2.53e-14 |
| 40923 | True | True | True | 1.36e-14 | 2.23e-14 | 2.58e-14 | 6.77e-15 | 8.93e-15 | 1.13e-14 |
| 41082 | True | True | True | 1.13e-15 | 3.91e-15 | 9.50e-15 | 1.60e-15 | 9.70e-15 | 5.18e-14 |
| 41159 | True | True | True | 1.84e-15 | 2.07e-15 | 2.76e-15 | 1.37e-15 | 4.44e-15 | 8.32e-15 |
| 43985 | True | True | True | 1.53e-14 | 1.57e-14 | 1.86e-14 | 3.82e-15 | 5.96e-15 | 7.58e-15 |

composition with accuracy comparable to the centralized algorithm, consistent with Theorem 3.2.

**Effectiveness of FedGCUR** We present the learning performance using the selected data and features in Table 4. The compared methods are abbreviated: the first word denotes the data selection method: Coreset (Sener & Savarese, 2018), Leverage Score (Lever.) (Larsen & Kolda, 2022), or Random (R.); while the second letter denotes the feature selection method: Variance (Var.) or Random. For reference, we also report the performance of using all data, though this is not intended for direct comparison. In the case of Fed-CPQR, all data points are used along with sampled features, whereas all other methods select the same number of rows and columns. Each experiment is repeated 10 times with different random seeds, and average results are reported. The best-performing method and all statistically comparable methods are highlighted in bold based on paired t-test of 0.05 significance level. Note that, when FedCPQR achieves the best performance, we instead highlight the second-best method and its comparable results, since FedCPQR uses all the data but the other methods only use a half.

The results show that FedCPQR and FedGCUR achieve the best or near-best performance in many settings. The trends are largely consistent between i.i.d. and non-i.i.d. splits. FedGCUR surpasses FedCPQR in several cases, suggesting that local row selection can remove harmful or redundant samples instead of merely reducing communication and computation. We note that performance on dataset 40923 is low for all methods because we fix the number of FedAvg communication rounds to 10 across datasets for fair comparison. Since FedCPQR and FedGCUR are label-free reconstruction methods rather than discriminative feature selectors, large downstream gains over task-specific baselines are not expected in every case; label-aware extensions are left for future work.

**Vision Benchmarks on Larger Feature Matrices** To further test whether the same selection mechanism remains useful beyond tabular OpenML tasks, we evaluate on four vision benchmarks: Fashion-MNIST (FMNIST), CIFAR-

10 (C10), CIFAR-100 (C100), and Tiny ImageNet (Tiny-Img). Fashion-MNIST contains 60,000 training images and 10,000 test images from 10 categories (Xiao et al., 2017). CIFAR-10 and CIFAR-100 each contain 50,000 training images and 10,000 test images from 10 and 100 classes, respectively (Krizhevsky, 2009). Tiny ImageNet is a 200-class subset of ImageNet with 500 training images, 50 validation images, and 50 test images per class (Le & Yang, 2015). For

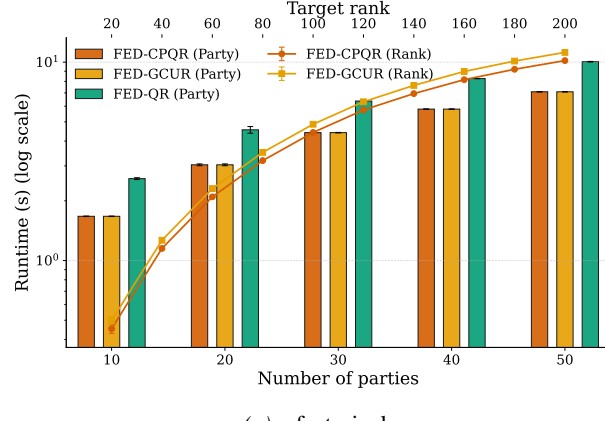

*(a)* mfeat-pixel

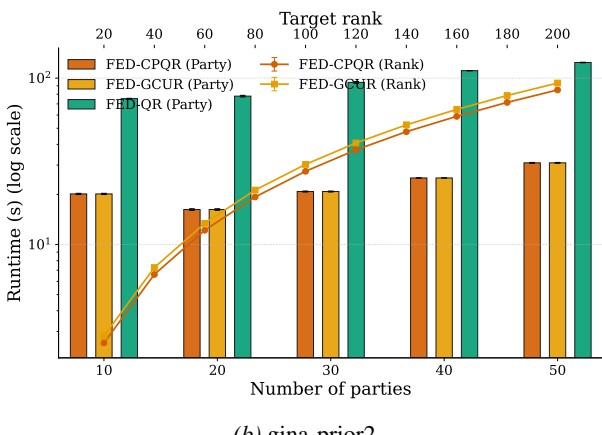

*(b)* gina_prior2

*Figure 1.* Log-scale running times (in seconds) of FedQR, Fed-CPQR, and FedGCUR with varying numbers of parties and ranks. Error bars indicating standard deviations of 5 runs.

*Table 4.* Task performance comparison of different data and feature selection methods on i.i.d. and non-i.i.d. split datasets. The mean and std. accuracy (%) are reported in the table. The best-performing method and all statistically comparable methods are highlighted in bold.

| ID | Rank | Oracle | FedCPQR | FedGCUR | Coreset.R | Coreset.Var. | Lever.R. | Lever.Var. | R.Variance |
|---|---|---|---|---|---|---|---|---|---|
| | | | | Performance on IID Split Data | | | | | |
| 1022 | 10 | 92.4±0.6 | **90.0±0.0** | 90.0±0.0 | 90.0±0.0 | 90.0±0.0 | 90.0±0.0 | 90.0±0.0 | 90.0±0.0 |
| | 50 | 92.4±0.6 | **90.0±0.0** | 90.0±0.0 | 90.0±0.0 | 90.0±0.0 | 90.0±0.0 | 90.0±0.0 | 90.0±0.0 |
| | 100 | 92.4±0.6 | **90.0±0.0** | **90.0±0.1** | **90.0±0.0** | **90.0±0.0** | **90.0±0.0** | **90.0±0.0** | **90.0±0.0** |
| 1041 | 10 | 58.8±1.3 | **11.1±1.6** | **10.4±2.0** | 10.5±0.0 | 10.5±0.0 | 10.5±0.0 | 10.6±0.1 | **11.1±0.0** |
| | 50 | 58.8±1.3 | **15.0±2.4** | **12.1±3.3** | 10.5±0.0 | 10.5±0.0 | 10.5±0.1 | 10.7±0.0 | 9.8±0.1 |
| | 100 | 58.8±1.3 | **15.4±3.4** | **17.5±2.4** | 11.5±0.6 | 11.5±0.6 | 14.8±0.8 | 10.1±0.1 | 12.2±0.6 |
| 40923 | 10 | 8.9±1.1 | **2.3±0.4** | **2.3±0.5** | **2.2±0.0** | **2.2±0.0** | **2.2±0.0** | **2.2±0.0** | **2.2±0.0** |
| | 50 | 8.9±1.1 | **2.6±0.5** | **2.2±0.5** | **2.2±0.0** | **2.2±0.0** | **2.2±0.0** | **2.2±0.0** | **2.2±0.0** |
| | 100 | 8.9±1.1 | **2.7±0.5** | **2.6±0.2** | 2.1±0.2 | 2.1±0.2 | **2.5±0.1** | 2.2±0.0 | 2.3±0.1 |
| 41082 | 10 | 60.0±0.8 | **26.4±3.8** | 20.9±7.0 | 17.2±0.7 | 17.2±0.7 | 19.1±0.5 | 16.3±0.6 | 24.1±1.0 |
| | 50 | 60.0±0.8 | 28.3±6.4 | **38.6±4.5** | 20.0±0.6 | 20.0±0.6 | 19.1±0.4 | 18.3±0.5 | 21.3±2.4 |
| | 100 | 60.0±0.8 | 42.2±4.7 | **46.9±2.8** | 25.7±1.2 | 25.7±1.2 | 37.2±1.1 | 29.9±0.6 | **46.2±0.8** |
| 41159 | 10 | 60.8±0.7 | **59.2±0.6** | 56.8±1.2 | **59.1±0.2** | **59.1±0.2** | 58.9±0.3 | **59.4±0.3** | 58.4±0.4 |
| | 50 | 60.8±0.7 | **59.0±0.6** | **58.8±0.5** | 58.2±0.5 | 58.2±0.5 | **58.7±0.2** | **58.6±0.3** | 56.2±0.7 |
| | 100 | 60.8±0.7 | 57.2±1.2 | **58.6±1.1** | 57.3±0.6 | 57.3±0.6 | **59.0±0.2** | 57.1±0.6 | 57.5±0.5 |
| 43985 | 10 | 78.7±1.2 | **15.8±2.7** | 12.7±1.6 | 14.3±0.6 | 14.3±0.6 | 12.2±0.6 | **16.9±0.8** | 15.4±0.8 |
| | 50 | 78.7±1.2 | **26.8±2.7** | **23.2±1.3** | 17.8±0.4 | 17.8±0.4 | 16.6±0.9 | 17.1±1.2 | 17.2±1.3 |
| | 100 | 78.7±1.2 | 35.7±1.8 | **37.9±2.0** | 24.4±0.8 | 24.4±0.8 | 21.2±0.5 | 21.5±0.9 | 24.8±2.0 |
| | | | | Performance on Non-IID Split Data | | | | | |
| 1022 | 10 | 91.7±0.2 | **90.0±0.0** | 90.0±0.0 | 90.0±0.0 | 90.0±0.0 | 90.0±0.0 | 90.0±0.0 | 90.0±0.0 |
| | 50 | 91.7±0.2 | **90.0±0.0** | 90.0±0.0 | 90.0±0.0 | 90.0±0.0 | 90.0±0.0 | 90.0±0.0 | 90.0±0.0 |
| | 100 | 91.7±0.2 | **90.0±0.0** | 90.0±0.0 | 90.0±0.0 | 90.0±0.0 | 90.0±0.0 | 90.0±0.0 | 90.0±0.0 |
| 1041 | 10 | 58.8±1.4 | **11.0±1.6** | **10.3±2.1** | 10.5±0.0 | 10.5±0.0 | 10.5±0.0 | 10.6±0.1 | **10.9±0.3** |
| | 50 | 58.8±1.4 | **15.8±2.1** | **12.8±2.7** | 10.5±0.0 | 10.5±0.0 | 10.5±0.1 | 10.7±0.0 | 9.8±0.1 |
| | 100 | 58.8±1.4 | **16.3±2.2** | **17.6±1.4** | 11.0±0.4 | 11.0±0.4 | 14.5±0.9 | 10.1±0.1 | 12.0±0.4 |
| 40923 | 10 | 9.3±0.2 | **2.1±0.3** | **2.3±0.4** | **2.2±0.0** | **2.2±0.0** | **2.2±0.0** | **2.2±0.0** | **2.2±0.0** |
| | 50 | 9.3±0.2 | **2.5±0.5** | **2.3±0.5** | **2.2±0.0** | **2.2±0.0** | **2.2±0.0** | **2.2±0.0** | **2.2±0.0** |
| | 100 | 9.3±0.2 | 2.5±0.3 | **2.8±0.3** | 2.3±0.2 | 2.3±0.2 | 2.6±0.1 | 2.2±0.1 | 2.3±0.1 |
| 41082 | 10 | 59.8±0.7 | 26.8±4.1 | **31.9±6.3** | 17.7±1.0 | 17.7±1.0 | 20.0±0.5 | 17.6±1.1 | 24.5±1.0 |
| | 50 | 59.8±0.7 | 29.7±5.7 | **39.3±1.5** | 22.1±2.6 | 22.1±2.6 | 18.6±0.2 | 17.9±0.7 | 20.5±1.0 |
| | 100 | 59.8±0.7 | 41.8±4.8 | 41.5±3.0 | 26.1±1.7 | 26.1±1.7 | 35.7±0.8 | 31.2±1.2 | **46.1±0.6** |
| 41159 | 10 | 60.7±0.6 | 59.1±0.6 | **60.0±0.1** | 59.1±0.3 | 59.1±0.3 | 58.6±0.2 | 59.1±0.3 | 58.5±0.5 |
| | 50 | 60.7±0.6 | 59.0±0.5 | **60.0±0.0** | 58.1±0.5 | 58.1±0.5 | 58.3±0.5 | 58.4±0.3 | 55.9±0.9 |
| | 100 | 60.7±0.6 | 57.2±1.2 | **60.0±0.0** | 57.1±0.6 | 57.1±0.6 | 58.8±0.2 | 56.8±0.6 | 57.5±0.6 |
| 43985 | 10 | 73.3±2.5 | **14.6±1.9** | 11.7±2.7 | 12.8±1.4 | 12.8±1.4 | 11.1±1.6 | **14.9±1.7** | **14.2±1.0** |
| | 50 | 73.3±2.5 | **25.3±2.7** | **16.9±2.1** | **17.7±2.7** | **17.7±2.7** | **16.8±1.3** | 16.3±2.3 | 16.5±2.1 |
| | 100 | 73.3±2.5 | **32.7±2.9** | **25.9±0.8** | 24.4±1.2 | 24.4±1.2 | 21.1±0.8 | 21.0±1.1 | 21.7±1.4 |

each dataset, selection is performed on 2048-dimensional ResNet-50 features extracted from an ImageNet-pretrained model and partitioned across 10 silos.

The federated largely setting follows the previous experiment. Each method is run over ten seeds with target rank $k = 1500$. Table 5 reports downstream federated test accuracy and absolute Frobenius CUR reconstruction errors. For reconstruction, each row-sampled method selects $\lfloor n_c/2 \rfloor$

rows from silo $c$ under a matched half-row budget. Given selected columns $J$ and rows $I$, we form $C = A[:, J]$, $R = A[I, :]$, $W = A[I, J]$, and reconstruct $\widehat{A} = CW^{\dagger}R$; per-silo errors apply the same coefficient $W^{\dagger}R$ locally.

Table 5 shows that FedCPQR remains a strong global feature-selection routine. On the reported i.i.d. downstream tasks, FedCPQR is the best or statistically comparable selected method on Fashion-MNIST, CIFAR-100, and Tiny

*Table 5.* Vision benchmark comparison of downstream accuracy and CUR reconstruction quality at rank $k = 1500$. Reconstruction error is the absolute Frobenius error. Accuracy is reported as mean $\pm$ std (%). WRE and SRE denote whole and per-silo reconstruction error, respectively, and reconstruction entries are reported in units of $10^2$, where lower is better. NIID denotes the non-i.i.d. split. The best selected method and statistically comparable methods are highlighted in bold.

| Data | Eval. | Oracle | FedCPQR | FedGCUR | Coreset.R | Coreset.Var. | Lever.R. | Lever.Var. | R.Variance |
|------|-------|--------|---------|---------|-----------|--------------|----------|------------|------------|
| FMNIST | Acc.-IID | 91.3±0.3 | **90.2±0.2** | 89.4±0.2 | 89.7±0.1 | **89.9±0.2** | **89.7±0.3** | 87.9±0.3 | **90.0±0.3** |
| C10 | Acc.-IID | 92.0±0.1 | 90.1±0.1 | 89.2±0.1 | **90.3±0.4** | **90.7±0.2** | **90.7±0.1** | 88.3±0.1 | **90.5±0.2** |
| C100 | Acc.-IID | 73.5±0.0 | **70.6±0.2** | 67.7±0.1 | **69.8±0.4** | 70.0±0.1 | 69.4±0.1 | 65.9±0.1 | 69.9±0.3 |
| Tiny-Img | Acc.-IID | 76.2±0.2 | **73.9±0.3** | 72.4±0.1 | 72.7±0.3 | 72.5±0.2 | 72.2±0.2 | 69.1±0.4 | 72.0±0.3 |
| FMNIST | Acc.-NIID | 90.3±0.1 | **89.1±0.3** | 87.2±0.8 | **89.2±0.1** | **89.3±0.1** | 88.4±0.1 | 87.2±0.3 | **89.2±0.2** |
| C10 | Acc.-NIID | 91.5±0.1 | 89.6±0.1 | 85.7±1.2 | 89.8±0.2 | **90.3±0.1** | **90.2±0.1** | 87.4±0.0 | **90.1±0.2** |
| C100 | Acc.-NIID | 72.4±0.3 | **69.3±0.4** | 63.0±0.2 | **68.3±0.1** | **69.2±0.5** | **68.3±0.4** | 64.7±0.1 | **68.9±0.4** |
| Tiny-Img | Acc.-NIID | 75.2±0.2 | **73.1±0.3** | 65.8±0.3 | **72.0±0.5** | **72.0±0.3** | 71.3±0.3 | 68.4±0.2 | 71.1±0.4 |
| FMNIST | WRE-IID | 0.0±0.0 | 29.2±0.0 | **29.1±0.0** | 39.3±0.3 | 42.7±0.2 | 38.7±0.2 | 36.1±0.0 | 42.4±0.5 |
| C10 | WRE-IID | 0.0±0.0 | 30.9±0.0 | **30.7±0.0** | 37.3±0.3 | 39.6±0.2 | 37.1±0.2 | 35.2±0.0 | 39.3±0.3 |
| C100 | WRE-IID | 0.0±0.0 | 30.9±0.0 | **30.8±0.0** | 37.0±0.3 | 39.1±0.3 | 37.0±0.1 | 35.1±0.0 | 38.5±0.3 |
| Tiny-Img | WRE-IID | 0.0±0.0 | 44.6±0.0 | **44.5±0.0** | 51.9±0.4 | 53.8±0.1 | 52.0±0.2 | 51.5±0.0 | 53.8±0.4 |
| FMNIST | WRE-NIID | 0.0±0.0 | 29.2±0.0 | **29.2±0.0** | 39.4±0.3 | 42.9±0.5 | 38.7±0.2 | 35.5±0.1 | 42.7±0.5 |
| C10 | WRE-NIID | 0.0±0.0 | 30.9±0.0 | **30.9±0.0** | 37.3±0.3 | 39.8±0.2 | 37.1±0.2 | 35.1±0.1 | 39.2±0.2 |
| C100 | WRE-NIID | 0.0±0.0 | **30.9±0.0** | 30.9±0.0 | 37.0±0.3 | 39.0±0.0 | 37.0±0.1 | 35.1±0.0 | 38.6±0.2 |
| Tiny-Img | WRE-NIID | 0.0±0.0 | **44.6±0.0** | 45.1±0.0 | 51.9±0.4 | 53.9±0.3 | 52.0±0.2 | 51.4±0.0 | 53.8±0.2 |
| FMNIST | SRE-IID | 0.0±0.0 | 9.2±0.0 | **9.2±0.0** | 12.4±0.1 | 13.5±0.1 | 12.2±0.1 | 11.4±0.0 | 13.4±0.2 |
| C10 | SRE-IID | 0.0±0.0 | 9.8±0.0 | **9.7±0.0** | 11.8±0.1 | 12.5±0.1 | 11.7±0.1 | 11.1±0.0 | 12.4±0.1 |
| C100 | SRE-IID | 0.0±0.0 | 9.8±0.0 | **9.7±0.0** | 11.7±0.1 | 12.4±0.1 | 11.7±0.1 | 11.1±0.0 | 12.2±0.1 |
| Tiny-Img | SRE-IID | 0.0±0.0 | 14.1±0.0 | **14.1±0.0** | 16.4±0.1 | 17.0±0.1 | 16.4±0.1 | 16.3±0.0 | 17.0±0.1 |
| FMNIST | SRE-NIID | 0.0±0.0 | 9.1±1.7 | **9.1±1.8** | 12.2±2.4 | 13.3±2.7 | 12.0±2.3 | 11.1±2.1 | 13.3±2.6 |
| C10 | SRE-NIID | 0.0±0.0 | 9.6±1.7 | **9.6±1.7** | 11.6±2.1 | 12.4±2.2 | 11.5±2.1 | 10.9±2.0 | 12.2±2.2 |
| C100 | SRE-NIID | 0.0±0.0 | **9.7±0.6** | 9.8±0.6 | 11.7±0.8 | 12.3±0.8 | 11.7±0.8 | 11.1±0.8 | 12.2±0.8 |
| Tiny-Img | SRE-NIID | 0.0±0.0 | **14.1±0.7** | 14.3±0.7 | 16.4±0.8 | 17.0±0.9 | 16.4±0.8 | 16.2±0.8 | 17.0±0.9 |

ImageNet, while CIFAR-10 has several marginally stronger task-specific row/feature baselines. On the non-i.i.d. downstream tasks, FedCPQR remains statistically comparable to the best selected method across the reported datasets.

The reconstruction results give a complementary matrix-approximation view. FedGCUR achieves the lowest selected-method error in 12 of the 16 whole-matrix and per-silo reconstruction cases. Under i.i.d. splits, FedGCUR is best on all four datasets for both whole-matrix and per-silo errors, including CIFAR-10. Under non-i.i.d. splits, FedGCUR remains best on CIFAR-10 and Fashion-MNIST, while FedCPQR is slightly better on CIFAR-100 and Tiny ImageNet. These results sharpen the empirical message: FedCPQR gives strong global feature compression, whereas FedGCUR provides a fully data-and-feature-compressed CUR representation with competitive downstream accuracy and consistently strong reconstruction quality.

**Efficiency of FedCPQR and FedGCUR** Figure 1 reports the log-scale running times of FedQR, FedCPQR, and FedGCUR when varying the number of parties and the target rank. Full results are deferred to Appendix. FedCPQR and FedGCUR remain efficient as either quantity grows and generally run faster than FedQR; FedGCUR is only slightly more expensive than FedCPQR because it adds local row selection after the global pivot computation.

## 5. Conclusion

We proposed FedGCUR, a unified federated data and feature selection framework based on generalized CUR decomposition. The core subroutine, FedCPQR, securely computes a global column-pivot order, while each silo performs local row selection. We proved exact equivalence between FedCPQR and centralized modified Gram-Schmidt CPQR under exact secure aggregation, and established per-silo and global reconstruction guarantees for FedGCUR. Experiments across OpenML tabular datasets and four vision benchmarks support the correctness, effectiveness, and efficiency of the framework.

## Acknowledgments

This work is supported by the Ministry of Education, Singapore, under its Academic Research Fund Tier 1 (RG101/24).

## Impact Statement

This paper advances efficient and privacy-aware federated learning through label-free data and feature selection. The proposed routines may reduce communication and computation costs in distributed learning systems, but deployment in sensitive domains should still include task-specific auditing, secure aggregation, differential-privacy accounting when needed, and governance of downstream models.

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

## A. Runtime Results

We provide the complete runtime results of six datasets here.

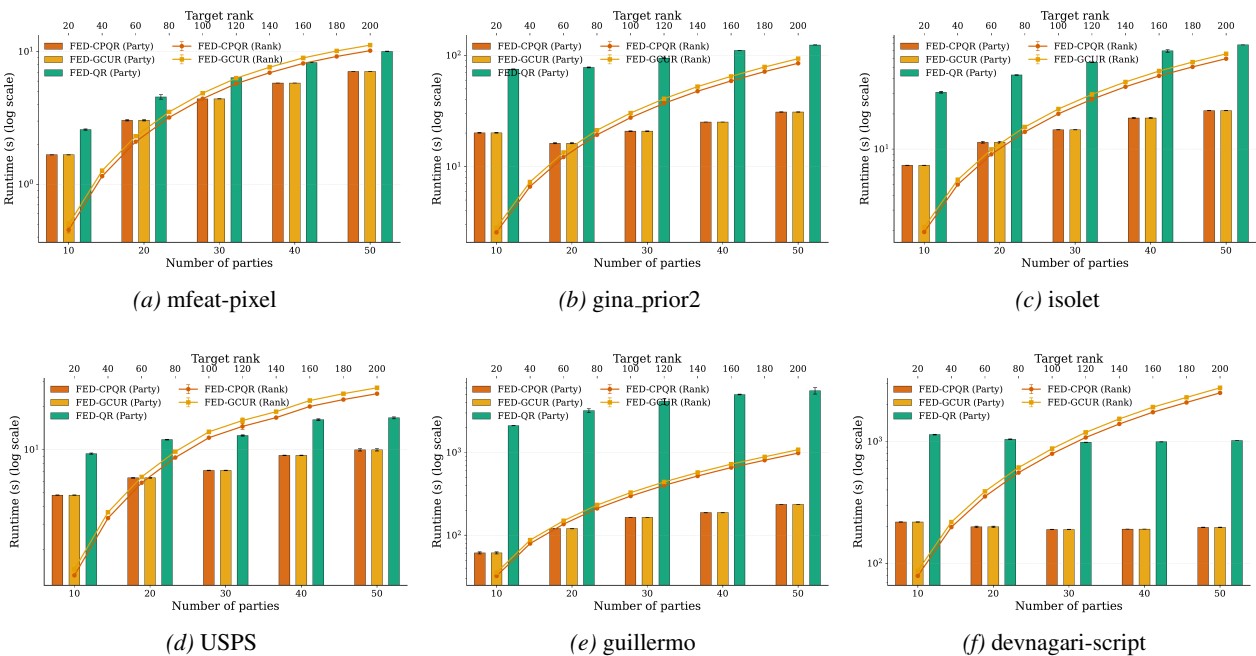

*Figure 2.* Log-scale running times (in seconds) of FedQR, FedCPQR, and FedGCUR with varying numbers of parties and ranks. Error bars indicating standard deviations of 5 runs.

## B. Proof of Theorem 3.2

We restate Theorem 3.2 first, and then present the proof.

**Theorem B.1.** *Assume exact arithmetic, exact secure aggregation of all required inner products and squared norms, deterministic tie-breaking for equal pivot norms, and nonzero pivot norms for the first $k$ iterations. Then the first $k$ pivot choices, triangular entries, local orthogonal blocks, and residual updates of FedCPQR are identical to those of centralized modified Gram–Schmidt based CPQR applied to $A = \left[A_1^\top, \ldots, A_s^\top\right]^\top$.*

*Proof.* We prove the claim by induction over the pivot step $i$.

At the beginning of the algorithm, each silo sets $U_c = A_c$. The centralized modified Gram-Schmidt CPQR algorithm starts from

$$U = A = \left[A_1^\top, \ldots, A_s^\top\right]^\top.$$

Therefore, the centralized working matrix is exactly the vertical stack of the local working matrices:

$$U = \left[U_1^\top, \ldots, U_s^\top\right]^\top.$$

The permutation matrix is initialized as $P = I_d$ in both algorithms, and the active candidate positions are initialized as $\{1, \ldots, d\}$ in both algorithms.

Assume that, before iteration $i$, the centralized working matrix satisfies

$$U = \left[U_1^\top, \ldots, U_s^\top\right]^\top,$$

and that the active candidate positions $\{i, \ldots, d\}$, the already selected pivots, and the previously computed entries of $R$ are identical in FedCPQR and centralized CPQR. For any remaining current position $j \in \{i, \ldots, d\}$, the squared column norm

used for pivoting in centralized CPQR is

$$\|U(:,j)\|_2^2 = \sum_{c=1}^{s} \|U_c(:,j)\|_2^2,$$

because $U(:,j)$ is the vertical concatenation of the local columns $\{U_c(:,j)\}_{c=1}^{s}$. FedCPQR computes the local terms $\|U_c(:,j)\|_2^2$ and obtains their exact sum through secure aggregation. Hence, FedCPQR and centralized CPQR assign the same pivot score to every active position $j \in \{i, \ldots, d\}$.

By the deterministic tie-breaking assumption, equal scores are resolved in the same way. Therefore, both algorithms select the same pivot column

$$p = \arg \max_{j \in \{i,\ldots,d\}} \|U(:,j)\|_2^2.$$

They consequently perform the same current-position column swap in $P$, in the working matrices, in the residual-norm vector, and in the already computed part of $R$.

After the pivot swap, the squared norm of the pivot column is again an additive quantity:

$$n_i = \|U(:,i)\|_2^2 = \sum_{c=1}^{s} \|U_c(:,i)\|_2^2.$$

FedCPQR obtains this exact value by secure aggregation. Since the pivot norm is nonzero by assumption, both algorithms set

$$R_{ii} = \sqrt{n_i}.$$

The centralized normalized vector is

$$Q(:,i) = \frac{U(:,i)}{\sqrt{n_i}} = \left[ \left( \frac{U_1(:,i)}{\sqrt{n_i}} \right)^{\top}, \ldots, \left( \frac{U_s(:,i)}{\sqrt{n_i}} \right)^{\top} \right]^{\top}.$$

FedCPQR sets $Q_c(:,i) = U_c(:,i)/\sqrt{n_i}$ at each silo. Thus, the stacked local vector $\left[ Q_1(:,i)^{\top}, \ldots, Q_s(:,i)^{\top} \right]^{\top}$ is exactly the centralized $Q(:,i)$.

For every remaining column $j > i$, centralized CPQR computes

$$R_{ij} = Q(:,i)^{\top} U(:,j).$$

Using the vertical block structure of $Q(:,i)$ and $U(:,j)$, this inner product decomposes as

$$Q(:,i)^{\top} U(:,j) = \sum_{c=1}^{s} Q_c(:,i)^{\top} U_c(:,j).$$

FedCPQR computes each local scalar $Q_c(:,i)^{\top} U_c(:,j)$ and obtains the exact global value through secure aggregation. Hence, both algorithms set the same $R_{ij}$.

Both algorithms then apply the same residual update. In centralized CPQR,

$$U(:,j) \leftarrow U(:,j) - Q(:,i) R_{ij}.$$

In FedCPQR, each silo applies

$$U_c(:,j) \leftarrow U_c(:,j) - Q_c(:,i) R_{ij}.$$

Stacking the local updates gives exactly the centralized update. Therefore, after the update, the centralized working matrix remains the vertical stack of the local working matrices.

We have shown that if the two algorithms agree before iteration $i$, then they select the same pivot, compute the same entries of $R$, construct the same stacked $Q(:,i)$, and produce the same residual working matrix after iteration $i$. Induction proves that the first $k$ pivot choices, triangular entries, local orthogonal blocks, and residual updates of FedCPQR are identical to those of centralized modified Gram-Schmidt based CPQR. □

## C. Proof of Theorem 3.3

We restate Theorem 3.3 for easier reading.

**Theorem C.1.** *Let $A \in \mathbb{R}^{n \times d}$ be partitioned by rows into $s$ parties, $A = \left[A_1^\top, \ldots, A_s^\top\right]^\top$, with $A_c \in \mathbb{R}^{n_c \times d}$. Let $D \in \mathbb{R}^{d \times k}$ be the column selection matrix returned by FedCPQR, let $C_c = A_c D$, let $T_c = S_c^\top A_c$, and let $\widehat{A}_c = C_c C_c^\dagger A_c T_c^\dagger T_c$ be the FedGCUR reconstruction for silo $c$. Define*

$$P_{C,c} := C_c C_c^\dagger, \qquad P_{R,c} := T_c^\dagger T_c, \qquad P_k := DD^\top,$$

$$P_{X,c} := D(A_c D)^\dagger A_c, \qquad \eta_c := \|I_d - P_{X,c}\|_2, \qquad \rho_{c,\xi} := \|P_{C,c} A_c (I_d - P_{R,c})\|_\xi.$$

*Then, for each silo $c$ and each $\xi \in \{2, F\}$,*

$$\|A_c - \widehat{A}_c\|_\xi \le \eta_c \|A_c(I_d - P_k)\|_\xi + \rho_{c,\xi}.$$

*Consequently, for $\widehat{A}_{\mathrm{loc}} = \left[\widehat{A}_1^\top, \ldots, \widehat{A}_s^\top\right]^\top$,*

$$\|A - \widehat{A}_{\mathrm{loc}}\|_2 \le \left(\sum_{c=1}^s \left[\eta_c \|A_c(I_d - P_k)\|_2 + \rho_{c,2}\right]^2\right)^{1/2},$$

*and*

$$\|A - \widehat{A}_{\mathrm{loc}}\|_F \le \left(\sum_{c=1}^s \left[\eta_c \|A_c(I_d - P_k)\|_F + \rho_{c,F}\right]^2\right)^{1/2}.$$

*Proof.* Fix a silo $c$. To simplify notation within the proof, write

$$A = A_c, \qquad C = C_c = A_c D, \qquad T = T_c, \qquad P_C = P_{C,c}, \qquad P_R = P_{R,c}, \qquad P_X = P_{X,c}.$$

By the definition of the FedGCUR middle factor,

$$\widehat{A}_c = C_c M_c T_c = C_c C_c^\dagger A_c T_c^\dagger T_c = P_C A P_R.$$

Therefore, the local error is

$$E_c = A - \widehat{A}_c = A - P_C A P_R.$$

Add and subtract $P_C A$:

$$E_c = A - P_C A + P_C A - P_C A P_R.$$

Factor the two terms:

$$E_c = (I - P_C)A + P_C A(I_d - P_R).$$

Taking either the spectral norm or the Frobenius norm and applying the triangle inequality gives

$$\|E_c\|_\xi \le \|(I - P_C)A\|_\xi + \|P_C A(I_d - P_R)\|_\xi, \qquad \xi \in \{2, F\}.$$

By definition, the second term is $\rho_{c,\xi}$:

$$\|P_C A(I_d - P_R)\|_\xi = \rho_{c,\xi}.$$

Recall that $P_k = DD^\top$ and $C = AD$. Since $P_C = CC^\dagger$ is the orthogonal projector onto the column space of $C$, it satisfies

$$P_C C = C.$$

Hence

$$(I - P_C)AP_k = (I - P_C)ADD^\top = (I - P_C)CD^\top = 0.$$

Using $I_d = P_k + (I_d - P_k)$ on the right of $(I - P_C)A$, we obtain

$$(I - P_C)A = (I - P_C)AP_k + (I - P_C)A(I_d - P_k) = (I - P_C)A(I_d - P_k).$$

Because $I - P_C$ is an orthogonal projector, its spectral norm is at most one. Thus,

$$\|(I - P_C)A\|_2 = \|(I - P_C)A(I_d - P_k)\|_2 \leq \|A(I_d - P_k)\|_2,$$

and, using $\|MB\|_F \leq \|M\|_2\|B\|_F$,

$$\|(I - P_C)A\|_F = \|(I - P_C)A(I_d - P_k)\|_F \leq \|A(I_d - P_k)\|_F.$$

It remains to connect this contractive bound with the displayed $\eta_c$ notation. First observe that

$$AP_X = ADC^\dagger A = CC^\dagger A = P_C A,$$

so $(I - P_C)A = A(I_d - P_X)$. The matrix

$$P_X = DC^\dagger A$$

is idempotent, because

$$P_X^2 = DC^\dagger ADC^\dagger A = DC^\dagger CC^\dagger A = DC^\dagger A = P_X,$$

where we used the Moore–Penrose identity $C^\dagger CC^\dagger = C^\dagger$. Therefore $I_d - P_X$ is also idempotent. If $I_d - P_X = 0$, then $(I - P_C)A = A(I_d - P_X) = 0$ and the desired bound is trivial. Otherwise, $I_d - P_X$ has eigenvalue 1, so

$$\eta_c = \|I_d - P_X\|_2 \geq 1.$$

Combining this with the two contractive inequalities above gives

$$\|(I - P_C)A\|_\xi \leq \eta_c\|A(I_d - P_k)\|_\xi, \qquad \xi \in \{2, F\}.$$

Substituting the column residual bound from Step 2 and the row residual identity from Step 1 yields

$$\|A_c - \widehat{A}_c\|_\xi \leq \eta_c\|A_c(I_d - P_k)\|_\xi + \rho_{c,\xi}, \qquad \xi \in \{2, F\}.$$

This proves the per-silo inequality.

For the stacked spectral norm, write

$$A - \widehat{A}_{\text{loc}} = \left[E_1^\top, \ldots, E_s^\top\right]^\top.$$

Then

$$\|A - \widehat{A}_{\text{loc}}\|_2^2 = \lambda_{\max}\left(\sum_{c=1}^s E_c^\top E_c\right) \leq \sum_{c=1}^s \lambda_{\max}(E_c^\top E_c) = \sum_{c=1}^s \|E_c\|_2^2.$$

Taking square roots and substituting the per-silo bound for $\|E_c\|_2$ gives the stated spectral-norm inequality.

For the Frobenius norm, the block rows add exactly:

$$\|A - \widehat{A}_{\text{loc}}\|_F^2 = \sum_{c=1}^s \|E_c\|_F^2.$$

Substituting the per-silo bound for $\|E_c\|_F$ and taking square roots gives the stated Frobenius-norm inequality. $\qquad\square$

## D. Privacy Protection

In the proposed FedCPQR method, the $R$ and $P$ are shared. When FedCPQR is run to completion, this implies the leakage of $G = A^\top A = PR^\top RP^\top$; when it is run only to rank $k$, it reveals the corresponding pivoted rank-$k$ Gram information. Under an honest-but-curious threat model, an adversary possessing these aggregate statistics can attempt to reconstruct $A$.

We simulate reconstruction attacks in which the adversary observes the released Gram information and tries to recover a data matrix consistent with it. The results are reported in Table 6. The table includes two OpenML tabular matrices and two image-feature matrices used only for the privacy stress test. Without DP noise, the attack succeeds because the clean Gram matrix is highly informative. After Gaussian noise is added using the calibration in Sec. 3.2, the attack fails across all tested settings. Even with a loose per-release budget of $\epsilon = 10$, the attack MSE increases substantially relative to the non-private baseline; with $\epsilon = 0.1$, the reconstruction is uninformative. The results are empirical and are meant to complement the formal DP accounting in Sec. 3.2.

*Table 6. Detailed reconstruction-attack results with $(\varepsilon, \delta)$-DP FedCPQR. The attack attempts to recover private data $A$ from the noisy Gram matrix $G = A^\top A$. Scalar contributions are clipped before aggregation and Gaussian noise is added to the released aggregate statistics.*

| Dataset | $\varepsilon$ | $\sigma$ | MSE ($\uparrow$) | Gram Error | Status |
|---|---|---|---|---|---|
| openml_1022 | $\infty$ | 0 | 0.0006 | 0.0000 | Success |
| | 0.1 | 48.4481 | 5.5199 | 3498.4065 | Failed |
| | 0.5 | 9.6896 | 1.0223 | 698.4952 | Failed |
| | 1.0 | 4.8448 | 0.4806 | 351.5789 | Failed |
| | 10.0 | 0.4845 | 0.0319 | 35.2871 | Failed |
| openml_1041 | $\infty$ | 0 | 0.0007 | 0.0000 | Success |
| | 0.1 | 48.4481 | 7.3933 | 1739.2357 | Failed |
| | 0.5 | 9.6896 | 1.3588 | 350.3930 | Failed |
| | 1.0 | 4.8448 | 0.6401 | 174.0046 | Failed |
| | 10.0 | 0.4845 | 0.0370 | 17.5390 | Failed |
| cifar10_resnet50 | $\infty$ | 0 | 0.0000 | 0.0000 | Success |
| | 0.1 | 48.4481 | 2.0404 | 34042.3945 | Failed |
| | 0.5 | 9.6896 | 0.3911 | 6801.8740 | Failed |
| | 1.0 | 4.8448 | 0.1908 | 3401.4888 | Failed |
| | 10.0 | 0.4845 | 0.0150 | 340.2355 | Failed |
| mnist_resnet50 | $\infty$ | 0 | 0.0001 | 0.0000 | Success |
| | 0.1 | 48.4481 | 2.0419 | 33631.0664 | Failed |
| | 0.5 | 9.6896 | 0.3918 | 6723.4473 | Failed |
| | 1.0 | 4.8448 | 0.1908 | 3363.5183 | Failed |
| | 10.0 | 0.4845 | 0.0150 | 336.1008 | Failed |

