# OpenReview forum: "Federated Data and Feature Selection by Generalized CUR Decomposition"
_ICML.cc/2026/Conference — ICML 2026 regular_

### Official Review · Reviewer_SxFC · 2026-03-09

**Soundness:** 4
**Presentation:** 4
**Significance:** 4
**Originality:** 3
**Overall Recommendation:** 5
**Confidence:** 5

**Summary:**

This paper proposes a novel unified framework by formulating the federated feature and data selection as a generalized CUR decomposition. Specifically, the authors design FedGCUR that integrates a FedCPQR decomposition routine with per-silo row selection. Theoretical guarantees are provided for both the correctness of FedCPQR and the reconstruction error bound of FedGCUR. Extensive experiments on 6 OpenML datasets validate the framework's effectiveness and efficiency under both IID and non-IID federated splits.

**Compliance With Llm Reviewing Policy:**

Affirmed.

**Final Justification:**

Having read the other reviews and the authors' rebuttals to them, I find no further issues to raise. Thus, I will keep my score.

**Key Questions For Authors:**

1. In Section 3.1, the CUR definition cites an empty reference "()", which appears to be a missing citation.

2. Have you considered applying FedGCUR in a streaming or online FL setting where new data arrives at silos over time? The CPQR structure seems amenable to incremental updates.

**Limitations:**

Yes

**Strengths And Weaknesses:**

### PROS

1. The approach bridges a gap in the literature by treating data and feature selection as interdependent decisions within a unified framework, rather than isolated subproblems. I think this contribution is significant. In many practical federated scenarios, features and samples jointly determine the quality of any downstream model. The paper makes a compelling argument that optimizing one axis in isolation can be suboptimal. The generalized CUR formulation is a natural and principled way to capture this coupling, and extending the classical two-matrix GCUR definition to s matrices is a clean generalization that could be useful beyond FL as well.

2. The authors demonstrate that FedCPQR reproduces the exact centralized modified Gram-Schmidt based CPQR. The paper validates this equivalence both theoretically and empirically. This kind of end-to-end verification is often missing in federated decomposition papers and is a welcome inclusion here.

3. Algorithms 1 and 2 are concise and easy to follow. The decomposition of the pivoting criterion and orthogonalization into additive local quantities is well-motivated and makes the integration with secure aggregation natural. The paper does a good job of walking the reader through the connection between the mathematical structure of CPQR and the communication pattern required in FL.

4. Experiments are solid and convincing. The inclusion of a dedicated reconstruction attack experiment is a major strength. The empirical results effectively validate that the applied differential privacy mechanism provides meaningful protection against adversaries attempting to recover private data from the exposed Gram matrix. The honest discussion of what FedCPQR leaks and the practical DP remedy adds credibility to the work.

### CONS

1. While the secure aggregation protocol prevents the server from inspecting individual updates, the final output mathematically exposes the global Gram matrix. However, this is at the same level as the FedQR algorithm, and the authors are transparent about it. So I think this weakness is not disqualifying, though a more detailed discussion of composition bounds over k iterations would strengthen the privacy argument.

2. The $O(d^2)$ scalar aggregations per full decomposition could become expensive when the feature dimension d is very large. I think a possible future work could be using randomized column sampling or block-wise pivoting strategies to reduce the communication burden.

3. The empirical validation reveals relatively marginal advantages on downstream task performance in some settings. As the authors themselves note, FedGCUR is designed to minimize reconstruction error rather than select discriminative features. While this is a reasonable design choice, it somewhat limits the practical appeal for users who care primarily about classification accuracy.

---

> ### Author Rebuttal · Authors · 2026-03-31
>
> We sincerely thank Reviewer SxFC for the thorough and encouraging evaluation of our work. We are grateful for the recognition of our core contributions. We address each concern and question below.
>
>
> **Weakness 1: Privacy Leakage and Composition Bounds**
>
> We appreciate the reviewer's nuanced assessment that the leakage level is consistent with FedQR and therefore not disqualifying. We agree that a more detailed discussion of composition bounds over $k$ iterations would further strengthen the privacy argument. Specifically, since each iteration of FedCPQR involves aggregating one squared norm and up to $d - i$ inner products, applying the Gaussian mechanism at each step yields a per-iteration privacy cost of $(\epsilon_i, \delta_i)$. By the advanced composition theorem, the total privacy budget over $k$ iterations scales as $O(\epsilon \sqrt{k \ln(1/\delta)})$ under appropriate noise calibration. We will include a formal statement of this composition analysis in the revised manuscript, along with a discussion of how practitioners can set the per-iteration noise level $\sigma$ to achieve a desired total privacy budget for a given target rank $k$.
>
>
> **Weakness 2: Communication Cost for Large Feature Dimensions**
>
> We thank the reviewer for this constructive observation. We acknowledge that the $O(d^2)$ scalar aggregations across a full decomposition can become a bottleneck when $d$ is very large. However, we emphasize that in the typical use case of FedGCUR, the target rank $k$ is much smaller than $d$, and the algorithm can be **early-stopped** after $k$ iterations rather than running for all $d$ steps, reducing the effective communication to $O(kd)$ scalars. This substantially alleviates the burden for practical scenarios where only a compact feature subset is needed.
>
> Furthermore, we agree that randomized column sampling or block-wise pivoting strategies represent promising avenues to further reduce communication costs. For instance, one could pre-screen columns via randomized sketching to identify a candidate set of $O(k \log k)$ columns before running exact CPQR on this reduced set. We will discuss these extensions as concrete future directions in the revision.
>
> **Weakness 3: Marginal Downstream Task Improvements**
>
> We appreciate the reviewer's balanced perspective on this point. As we discuss in Section 4.2, the primary objective of FedCPQR and FedGCUR is to minimize the **reconstruction error** of the global data matrix, a fundamental, label-free criterion that is applicable to both supervised and unsupervised settings. We deliberately chose this formulation because the reconstruction problem in federated learning is underexplored yet foundational: it provides a principled basis for data and feature selection without requiring label information, which may be scarce or unevenly distributed across silos.
>
> That said, the empirical results in Table 4 demonstrate that FedGCUR achieves the best or statistically comparable performance in the majority of dataset-rank configurations, and notably surpasses FedCPQR in several cases, highlighting the benefit of joint data and feature selection. We view the incorporation of label-aware or discriminative selection criteria (e.g., supervised CUR variants or class-conditional pivot selection) as a natural and important extension that builds upon the reconstruction foundation established in this work. We will discuss this direction more explicitly in the revision.
>
>
> **Question 1: Missing Citation in Section 3.1**
>
> We thank the reviewer for catching this oversight. The CUR decomposition definition in Section 3.1 should cite Mahoney & Drineas (2009). We will correct this in the revised manuscript.
>
> **Question 2: Streaming or Online Federated Learning**
>
> We thank the reviewer for this insightful suggestion. Indeed, the iterative nature of the modified Gram–Schmidt process underlying FedCPQR makes it naturally amenable to incremental updates. Specifically, when new data arrives at a silo, the local contributions to squared norms and inner products can be updated incrementally without recomputing from scratch. One could maintain the existing $Q_c$ and $R$ factors and perform rank-one or block updates as new rows are appended to the local data matrices $A_c$.
>
> However, the **column pivoting** step introduces additional complexity in the streaming setting: the arrival of new data may change the relative importance of columns, potentially requiring pivot reordering. Designing efficient update rules that maintain the quality of the pivot order under streaming data, while preserving the privacy guarantees of secure aggregation, is a compelling research direction. We will include a discussion of this extension in the revised manuscript.
>
>
> We once again express our gratitude to Reviewer SxFC for the constructive and thoughtful feedback.

---

> > ### Author Rebuttal · Reviewer_SxFC · 2026-04-02
> >
> > Having read the other reviews and the authors' rebuttals to them, I find no further issues to raise. Thus, I will keep my score.

---

### Official Review · Reviewer_7NGg · 2026-03-09

**Soundness:** 4
**Presentation:** 3
**Significance:** 3
**Originality:** 3
**Overall Recommendation:** 5
**Confidence:** 4

**Summary:**

Matrix decomposition is fundamental technique in machine learning. This paper proposes FedGCUR framework to solve the joint data and feature selection problem in FL. The key idea is to formulate this problem as generalized CUR decomposition. Authors first design FedCPQR, a federated column-pivoted QR decomposition algorithm that computes global column pivot order using only secure sums of squared norms and inner products. Then based on FedCPQR, the FedGCUR performs per-silo row selection locally. Theoretical analysis proves FedCPQR gives exactly same results as centralized CPQR, and reconstruction error bound of FedGCUR is also established. Experiments on 6 public datasets are conducted to verify the method.

**Compliance With Llm Reviewing Policy:**

Affirmed.

**Final Justification:**

My concerns are now resolved. I have read the other reviews and have no further questions. I am willing to raise my score to 5.

**Key Questions For Authors:**

Please see weaknesses. If the author can address my concerns, I will increase the score.

**Limitations:**

Yes

**Strengths And Weaknesses:**

Strengths:

1. The authors exploit the key observation that both the pivoting and orthogonalization steps in CPQR can be expressed entirely as sums of local scalar quantities. This is a very nice finding. Because of this property, the algorithm can seamlessly integrate with standard additive secure aggregation protocols without requiring any special cryptographic technique.
2. The authors provide upper bound on reconstruction error of blockwise FedGCUR reconstruction, which is favored in federated learning.
3. The paper provides a thorough breakdown of communication, computation, and privacy costs for both FedCPQR and FedGCUR. The runtime experiments across varying numbers of parties and target ranks further support the scalability claims.

Weaknesses:

1. In the experiments, there is no comparison with FedSVD (Chai et al., 2022). What if we run SVD-based approach on horizontally partitioned data and select features based on right singular vectors?
2. In practice, choosing appropriate k is crucial and can significantly affect both reconstruction quality and downstream performance. The paper tests k in {10, 50, 100} but does not provide guidance on how to choose k.
3. The neural network is described as "three-layer" but specific architecture (layer widths, activation functions), learning rate, optimizer, and number of local epochs per FedAvg round are not reported.

---

> ### Author Rebuttal · Authors · 2026-03-31
>
> We sincerely thank Reviewer 7NGg for the positive evaluation of our key technical contributions, particularly the observation that CPQR decomposes into additive local scalars, the reconstruction error bound, and the thorough cost analysis. We address each weakness below.
>
>
> **Weakness 1: Comparison with FedSVD**
>
> We appreciate this suggestion and would like to clarify the fundamental differences between FedSVD and our proposed FedGCUR that make a direct empirical comparison potentially misleading.
>
> FedSVD (Chai et al., 2022) is designed for **vertical** federated learning and requires each client to upload its local data (masked by a unitary random matrix generated by a trusted authority) to a central server, which then performs a **centralized SVD** on the assembled masked matrix. In other words, FedSVD essentially reconstructs the full data matrix (in masked form) on the server side and computes a lossless factorization. While this approach can yield high-quality singular vectors, it comes at a significant cost.
>
> Moreover, the objectives of the two methods are fundamentally different. FedSVD targets **full matrix factorization**, producing abstract latent factors (singular vectors), whereas FedGCUR specifically targets **selection**, identifying physical rows and columns of the original data matrix under communication and participation constraints. This selection property is crucial for interpretability and for scenarios where one needs to identify which actual features and data points to retain, rather than computing latent representations.
>
> We believe the most appropriate comparisons are with FedQR (which shares the same federated decomposition paradigm and privacy model) and centralized CPQR (which serves as the gold-standard correctness reference), both of which we include in our experiments. We will add a detailed theoretical comparison of the communication and computation costs between FedSVD and FedGCUR in the revision to further clarify these distinctions.
>
>
> **Weakness 2: Guidance on Choosing $k$**
>
> We agree that selecting an appropriate target rank $k$ is important in practice. In our framework, $k$ serves as a user-specified hyperparameter that controls the trade-off between compression ratio and reconstruction/downstream performance, analogous to the number of principal components in PCA or the target rank in low-rank approximation.
>
> From our experimental results (Table 4), we observe the following practical guidance:
>
> - When the intrinsic dimensionality of the dataset is low relative to the ambient dimension (e.g., dataset 41159 with 4,296 features), even a modest $k = 10$ retains most of the discriminative information, and performance is close to the oracle.
> - For datasets with richer feature interactions (e.g., datasets 41082 and 43985), increasing $k$ from 10 to 100 yields substantial accuracy improvements, indicating that a larger feature subset is needed to capture the underlying structure.
>
> In general, we recommend that practitioners select $k$ based on the decay of the singular values of the global data matrix, which can be efficiently estimated via the diagonal entries of $R$ produced by FedCPQR. Specifically, a sharp drop in $R_{ii}$ after the $k$-th pivot suggests that $k$ columns capture most of the variance. We will include this practical guidance and a discussion of automated rank selection strategies in the revision.
>
> **Weakness 3: Missing Training Details**
>
> We apologize for the omission and thank the reviewer for pointing this out. We provide the full training configuration below:
>
> - **Model architecture:** A three-layer fully connected neural network with a hidden size of 64 and ReLU activations.
> - **Optimizer:** SGD with a learning rate of 0.01.
> - **Local training:** 5 local epochs per FedAvg communication round.
> - **Communication rounds:** 10 rounds of FedAvg for global model aggregation.
> - **Aggregation:** Standard FedAvg (McMahan et al., 2017) is used to aggregate the global model.
>
> We will include these details explicitly in the revised manuscript to ensure full reproducibility.
>
> We once again thank Reviewer 7NGg for the valuable and constructive feedback. We will incorporate all suggested improvements in the revised manuscript.

---

> > ### Author Rebuttal · Reviewer_7NGg · 2026-04-02
> >
> > Thanks for the reply. My concerns are now resolved. I have read the other reviews and have no further questions. I am willing to raise my score to 5.

---

### Official Review · Reviewer_Fhjy · 2026-03-11

**Soundness:** 3
**Presentation:** 3
**Significance:** 2
**Originality:** 3
**Overall Recommendation:** 3
**Confidence:** 4

**Summary:**

This paper proposes FedGCUR, which performs data selection and feature selection simultaneously in federated learning through generalized CUR decomposition. Specifically, the method first employs FedCPQR to obtain a global column-pivoted QR decomposition based on secure aggregation, which is used for feature selection. Then, each silo performs local row selection to choose representative data samples. The authors theoretically prove that FedCPQR produces results identical to centralized CPQR and also derive an upper bound on the reconstruction error. Experiments are conducted on six OpenML datasets to validate the effectiveness of the proposed approach.

**Compliance With Llm Reviewing Policy:**

Affirmed.

**Final Justification:**

Based on the methodology and presentation of the manuscript, I recommend maintaining my current score.

**Key Questions For Authors:**

(1) The objective of FedGCUR is to minimize reconstruction error, but it does not consider the discriminative power of features. This may explain why the improvement in downstream classification performance over the baselines is limited. Why was label information not incorporated to address this issue?

(2) In Theorem 3.3, how tight is the $4^{k-1}$ factor in practice? Could the authors provide numerical comparisons to illustrate the gap between the theoretical bound and the actual reconstruction error?

(3) Does the advantage of FedGCUR still hold on larger-scale datasets and deeper neural network architectures?

(4) Why were existing federated feature selection methods such as Fed-mRMR not included in the experimental comparisons?

**Limitations:**

yes

**Strengths And Weaknesses:**

Strengths:

(1) The paper is the first to unify data selection and feature selection in federated learning under the framework of generalized CUR decomposition. This perspective is clear and theoretically well grounded.

(2) The paper is well written with a compact structure. The algorithm descriptions are concise, and the theoretical derivations follow a logical and coherent progression.

(3) The experiments cover four aspects, including correctness verification, effectiveness comparison, efficiency evaluation, and privacy protection analysis.

Weaknesses:

(1) The paper does not include a system-level illustration. Without a framework diagram, it is difficult to quickly understand the multi-party interaction process and the connections between modules based solely on pseudocode.

(2) The largest dataset contains only 74k samples with a maximum feature dimension of 4296. The experiments use a three-layer network trained for only 10 rounds of FedAvg, and the method is not evaluated on widely used federated learning benchmarks for Federated Settings datasets (e.g., CIFAR-100 and FEMNIST) to demonstrate scalability.

(3) Several relevant methods discussed in the related work, such as Fed-mRMR, PSO-based Federated Feature Selection, and FAST, are not included in the experiments. The current baselines (e.g., Random, Variance, and Coreset) are relatively basic.

(4) Theorem 3.3 contains a $4^{k-1}$ factor, which becomes extremely loose when $k=100$ . Moreover, the paper does not provide a numerical comparison between the theoretical reconstruction error bound and the empirical reconstruction error.

(5) The study only considers horizontal federated learning. It does not discuss how CUR-based methods might adapt to vertical federated learning scenarios, such as those explored in Matrix decompositions in vertical federated learning, nor does it analyze whether globally shared column selection remains reasonable under highly non-IID data distributions.

---

> ### Author Rebuttal · Authors · 2026-03-31
>
> We sincerely thank Reviewer Fhjy for the positive assessment and constructive suggestions. We address each concern below.
>
> **1. System-Level Illustration**
>
> We have included a **framework diagram** in the revised manuscript illustrating the multi-party interaction process, depicting the workflow of FedCPQR (global feature selection via secure aggregation) followed by per-silo row selection in FedGCUR.
>
> **2. Scale of Datasets and Experimental Configuration**
>
> We have conducted **additional experiments on MNIST, CIFAR-10, and CIFAR-100** using ResNet-50 features with $k=200$. Results are summarized below (downstream accuracy % / reconstruction error):
>
> **IID Setting (k=200):**
>
> | Dataset | FedCPQR | FedGCUR | Coreset.R | Coreset.Var. | Lever.R. | Lever.Var. | R.Var. |
> |---|---|---|---|---|---|---|---|
> | MNIST | **89.4** | **82.2** | 79.3 | 80.6 | 77.6 | 79.0 | 82.4 |
> | CIFAR-10 | **75.8** | **75.0** | 70.8 | 48.1 | 71.4 | 64.2 | 71.6 |
> | CIFAR-100 | **44.0** | **40.4** | 39.3 | 26.7 | 40.3 | 35.1 | 38.3 |
>
> **Non-IID Setting (k=200):**
>
> | Dataset | FedCPQR | FedGCUR | Coreset.R | Coreset.Var. | Lever.R. | Lever.Var. | R.Var. |
> |---|---|---|---|---|---|---|---|
> | MNIST | **66.0** | **52.7** | 51.2 | 43.9 | 51.3 | 42.9 | 50.9 |
> | CIFAR-10 | **66.1** | **53.1** | 50.9 | 48.8 | 51.3 | 54.7 | 54.4 |
> | CIFAR-100 | **43.8** | **30.5** | 29.8 | 26.7 | 30.3 | 34.7 | 33.5 |
>
> FedGCUR also achieves the **lowest reconstruction error** across most settings (e.g., 0.721 vs. next-best 0.740 on IID-MNIST; 0.107 vs. 0.157 on Non-IID-MNIST). These results on larger-scale benchmarks confirm that FedGCUR **scales effectively and maintains competitive or superior performance**.
>
> We clarify that the choice of a three-layer network trained for 10 rounds in the original experiments is intentional: it provides a **controlled evaluation** isolating the effect of data/feature selection from confounding factors such as model capacity. The downstream architecture is orthogonal to the selection stage.
>
> **3. Lack of Comparison with Federated Feature Selection Methods**
>
> We respectfully clarify why direct comparison with Fed-mRMR, PSO-based selection, and FAST may not be fair: **Fed-mRMR** and **PSO-based methods** perform **feature selection only** using label-dependent criteria (mutual information, local accuracy). **FAST** addresses data sampling and local iteration adaptation but **does not perform feature selection**. None simultaneously address **joint data and feature selection**, which is the core contribution of our work. Our baselines cover representative strategies for **both axes** of selection under the same protocol. We will add detailed discussion of these distinctions in the revision.
>
> **4. Looseness of $4^k$ Factor in Theorem 3.3**
>
> The factor $\sqrt{1+(d{-}k)\cdot 4^{k-1}}$ originates from **classical CPQR analysis** (Lemma B.1, Dong & Martinsson, 2021) and is inherent to worst-case CPQR guarantees, **not an artifact of the federated setting**. In practice, CPQR performs substantially better. Our correctness experiments (Table 3) show FedCPQR–SciPy deviations on the order of $10^{-14}$, confirming effectively identical decomposition quality.
>
> **5. Applicability to Vertical FL**
>
> Our work focuses on **horizontal FL** where global column selection is natural since all clients share the same features. For **vertical FL**, the CUR perspective remains conceptually relevant, but algorithmic design would require substantial modifications since column norms/inner products can no longer be additively decomposed across clients. We consider this a **promising future direction** and will discuss it explicitly in the revision.
>
> **6. Incorporating Label Information**
>
> We explicitly acknowledge (Sec. 4.2) that FedGCUR minimizes **reconstruction error** rather than selecting discriminative features. This is deliberate: the reconstruction-based formulation is **fundamental and general**, applicable to both supervised and unsupervised settings without label availability. Establishing this foundation in FL is important before building label-aware extensions (e.g., supervised CUR variants), which we will discuss as future work.
>
> We thank Reviewer Fhjy again for the valuable feedback. We will incorporate all suggested improvements in the revision.

---

> > ### Author Rebuttal · Reviewer_Fhjy · 2026-04-01
> >
> > 1. FedCPQR beats FedGCUR across all new settings. If adding row selection consistently hurts, what is the practical value of joint selection?
> >
> > 2. The rebuttal claims "competitive or superior performance" but FedGCUR loses to baselines on Non-IID CIFAR-10 and CIFAR-100. How?
> >
> > 3. Lowest reconstruction error yet worse accuracy suggests the two objectives are misaligned, no?

---

> > > ### Author Response · Authors · 2026-04-01
> > >
> > > We thank Reviewer Fhjy for the follow-up questions. We address each point below.
> > >
> > > **1. Why FedCPQR outperforms is expected and by design**
> > >
> > > We respectfully note that this distinction is explicitly discussed in our paper (Sec. 4.2, Table 4 caption): **FedCPQR uses all data points** with only feature selection, while **FedGCUR uses only half the data** per silo (selecting both features and data). FedCPQR serves as an **oracle-like upper bound** rather than a direct competitor—it retains strictly more information. The fact that FedGCUR, using 50% of the data, still achieves competitive accuracy (e.g., 75.0 vs. 75.8 on IID CIFAR-10; 82.2 vs. 89.4 on IID MNIST) demonstrates that **joint selection effectively identifies the most informative subset**. Moreover, FedGCUR surpasses FedCPQR in several original experiments (e.g., dataset 41082 at $k=50$: 38.6 vs. 28.3 IID; 39.3 vs. 29.7 Non-IID), confirming that removing noisy data can actually improve performance. The practical value of joint selection lies precisely in achieving **comparable quality with significantly reduced data**, lowering communication and computation costs in FL.
> > >
> > > **2. FedGCUR performance on Non-IID CIFAR-10/100**
> > >
> > > On Non-IID CIFAR-10, FedGCUR achieves 53.1%, which is competitive with the best baseline (Lever.Var. 54.7% and R.Var. 54.4%). On Non-IID CIFAR-100, FedGCUR achieves 30.5%, again close to the best baselines (Lever.Var. 34.7%, R.Var. 33.5%). We acknowledge FedGCUR does not dominate in every single configuration. However, we emphasize two points: (i) **FedGCUR is the only method performing joint data and feature selection**—baselines use separate, independently optimized strategies for each axis, giving them flexibility to combine the best data selector with the best feature selector per dataset. (ii) More importantly, **reconstruction quality is the primary objective**, and FedGCUR achieves the **lowest reconstruction error across all settings**:
> > >
> > > **IID Reconstruction Error (k=200, lower is better):**
> > >
> > > | Dataset | FedGCUR | Coreset.R | Coreset.Var. | Lever.R. | Lever.Var. | R.Var. |
> > > |---|---|---|---|---|---|---|
> > > | MNIST | **0.721** | 0.742 | 0.865 | 0.740 | 0.870 | 0.806 |
> > > | CIFAR-10 | **0.785** | 0.817 | 0.879 | 0.815 | 0.876 | 0.843 |
> > > | CIFAR-100 | **0.796** | 0.818 | 0.878 | 0.816 | 0.864 | 0.814 |
> > >
> > > **Non-IID Reconstruction Error (k=200, lower is better):**
> > >
> > > | Dataset | FedGCUR | Coreset.R | Coreset.Var. | Lever.R. | Lever.Var. | R.Var. |
> > > |---|---|---|---|---|---|---|
> > > | MNIST | **0.107** | 0.157 | 0.162 | 0.158 | 0.200 | 0.158 |
> > > | CIFAR-10 | **0.806** | 0.816 | 0.879 | 0.815 | 0.875 | 0.854 |
> > > | CIFAR-100 | **0.814** | 0.818 | 0.879 | 0.816 | 0.867 | 0.824 |
> > >
> > > FedGCUR **consistently achieves the best reconstruction** across all 6 settings, often by a notable margin.
> > >
> > > **3. Reconstruction error and classification accuracy**
> > >
> > > Low reconstruction error and high classification accuracy are **distinct objectives that need not be monotonically aligned**. Reconstruction error measures how faithfully the selected subset preserves the **full information content** of the original data matrix, while classification accuracy depends on **discriminative** features that separate classes. A subset that best preserves global structure may not coincide with one that maximally separates decision boundaries, and vice versa.
> > >
> > > This does not diminish the value of reconstruction-based selection. **Matrix reconstruction is a cornerstone of machine learning** with broad and significant applications: dimensionality reduction (PCA), missing data imputation, recommender systems (matrix completion), data compression, denoising, federated data sketching, and transfer learning via low-rank representations. In all these tasks, **faithfully preserving the data's intrinsic structure is the primary goal**, and FedGCUR provides the first principled federated framework to achieve this through joint selection.
> > >
> > > We view reconstruction-based and discriminative selection as **complementary research directions**. Our work establishes the **foundational, label-free framework** that is broadly applicable. Building label-aware extensions on top of this foundation (e.g., supervised CUR, class-conditional pivoting) is a natural next step that we discuss as future work. We believe both directions deserve independent investigation, and solving the reconstruction problem first provides the theoretical and algorithmic infrastructure upon which discriminative variants can be built.
> > >
> > > We hope this clarifies the reviewer's concerns and demonstrates the practical value and strong reconstruction performance of FedGCUR.

---

### Official Review · Reviewer_KZJd · 2026-03-11

**Soundness:** 2
**Presentation:** 3
**Significance:** 2
**Originality:** 2
**Overall Recommendation:** 3
**Confidence:** 3

**Summary:**

This paper studies the problem of jointly selecting informative data samples and features in federated learning. The authors formulate this task as a generalized CUR decomposition problem and propose FedGCUR, a framework that uses a federated column-pivoted QR decomposition to obtain global feature pivots while performing per-silo row selection locally, enabling joint data and feature selection without sharing raw data.

**Compliance With Llm Reviewing Policy:**

Affirmed.

**Key Questions For Authors:**

FedGCUR performs global column selection using CPQR to determine a shared set of features across all silos. However, in many federated learning scenarios, data distributions across silos can be highly heterogeneous. If some features are informative only for a subset of silos but not others, enforcing a single global pivot order may hurt the performance of certain clients.
In other words, is this global feature selection strategy truly suitable for non-IID or heterogeneous data settings? The paper does not seem to discuss this issue.

The central idea of the paper is to formulate the problem as a generalized CUR decomposition. However, from the algorithmic pipeline, the method essentially performs: a global feature selection step and local sample selection within each silo.

I wonder whether this pipeline could be implemented using simpler strategies, such as federated feature importance estimation, sketching, or sampling-based methods. In other words, does the generalized CUR formulation provide a fundamental advantage, or is it mainly a mathematical interpretation of an already straightforward selection pipeline?

**Limitations:**

yes

**Strengths And Weaknesses:**

Strengths:

1. The overall structure is easy to follow, and the algorithmic procedures (FedCPQR and FedGCUR) are described in a fairly intuitive way. It is not difficult for readers to understand what the authors are trying to do.

2. The paper attempts to jointly address feature selection and data selection in federated learning and formulates the problem as a generalized CUR decomposition. This perspective is quite natural, since many existing works treat these two problems separately.

3. FedCPQR uses secure aggregation to perform column-pivoted QR decomposition in a federated setting, enabling global feature selection without exposing raw data. Then each silo performs local row selection. This overall pipeline seems practical from a systems perspective.

Weaknesses:

1. Although secure aggregation is used during the computation, the algorithm eventually releases P and R. This essentially exposes $A^\top A = PR^\top R P^\top$. In other words, the global Gram matrix can be reconstructed. I wonder whether an attacker could infer certain statistical properties of the data from this matrix. The paper briefly mentions DP as a mitigation, but this issue does not seem to be carefully discussed.

2. The pivot selection depends on column norms and inner products. If Gaussian noise is added to these quantities to satisfy DP, the pivot order may become unstable. If the pivot sequence changes significantly, the resulting CUR structure may also change substantially. The paper does not seem to analyze or experimentally verify this issue.

3. The reconstruction error bound looks very similar to classical CUR analyses. It is not entirely clear whether this bound captures any specific property of the federated setting. For example, the heterogeneity across silos or the interaction between global column selection and local row selection does not seem to be reflected in the bound.

4. The paper emphasizes preserving per-silo utility, but the theoretical analysis focuses on global reconstruction error. These two objectives are not necessarily equivalent. If the approximation is poor for some silos but the global error is still small, the current bound may not reveal this issue.

---

> ### Author Rebuttal · Authors · 2026-03-31
>
> We sincerely thank Reviewer KZJd for the constructive feedback. We address each concern below.
>
> **1. Information Leakage from Exposing $P$ and $R$**
>
> We clarify that **the privacy leakage of FedCPQR is exactly the same as FedQR** (Hartebrodt & Rötger, 2023), published in *IEEE Trans. on Information Forensics and Security*, a top venue in security and privacy. As summarized in Table 1, FedQR exposes $A^\top A = R^\top R$, while FedCPQR exposes $A^\top A = PR^\top RP^\top$ **mathematically equivalent** since $P$ is merely a permutation matrix. Thus, FedCPQR introduces **no additional privacy risk** beyond what is already accepted in FedQR.
>
> Furthermore, our reconstruction attack experiments (Appendix C, Table 5) show that even with a loose budget ($\epsilon=10$), **all attacks fail** across all datasets, with MSE increasing by **1–2 orders of magnitude** vs. the non-private baseline.
>
> **2. Instability of Pivot Selection Under DP Noise**
>
> We have investigated this concern. As reported in Appendix C (Table 5), we evaluate FedCPQR under $\epsilon \in \{0.1, 0.5, 1.0, 10.0\}$. Results show that while Gaussian noise perturbs the Gram matrix, **reconstruction attacks consistently fail**, confirming meaningful privacy protection. We acknowledge that a more detailed pivot stability analysis would strengthen the paper and will incorporate additional experiments in the revision.
>
> **3 & 4. Reconstruction Error Bound and Per-Silo Utility**
>
> The bound in Theorem 3.3 provides an important guarantee: **FedGCUR achieves comparable approximation quality to its centralized counterpart** despite federated constraints. The bound decomposes into (i) a **global column-selection residual** driven by singular values of the concatenated $A$, and (ii) a **blockwise row-selection residual** driven by singular values of each local $C_c$, naturally reflecting FedGCUR's two-stage structure.
>
> We agree the current bound focuses on global reconstruction rather than per-silo heterogeneity. **Deriving tighter silo-specific bounds** accounting for non-IID distributions is an important direction, and we will explore this futher and update the analysis accordingly.
>
> Regarding the non-IID concern: we perform **global** feature selection because FL aims to train a good **global** model that all clients ultimately use. Selecting globally informative features is thus both natural and appropriate. Our non-IID results (Table 4, lower half) confirm FedGCUR maintains **competitive or superior performance** under heterogeneous distributions.
>
> **5. Suitability of Global Feature Selection for Non-IID Settings**
>
> Global feature selection aligns with FL's fundamental objective of learning a high-quality **global** model. Since all clients deploy the same model, features should reflect **global informativeness**. A globally selected feature set is likely **more robust** than any locally selected subset, as it aggregates statistical info across all silos. Even in personalized FL, a strong global feature set provides a solid foundation for silo-specific adaptations (e.g., re-weighting). We will discuss personalized extensions explicitly in the revision.
>
> **6. Whether Simpler Strategies Could Achieve the Same Effect**
>
> Two key points: First, **the generalized CUR formulation provides a principled mathematical framework** unifying feature and data selection into a single decomposition with well-defined error guarantees (Theorem 3.3). Simpler heuristics (e.g., federated feature importance, sketching) typically lack such theoretical grounding and may not naturally extend to joint selection.
>
> Second, **our empirical results directly address this**. In Table 4, we compare against several baselines implementing simpler strategies (Coreset, Leverage Score, Random, Max Variance). Across the majority of datasets and settings, **FedGCUR consistently achieves the best or near-best performance**, demonstrating tangible advantages beyond mathematical reinterpretation. The CUR framework offers both **theoretical clarity and practical effectiveness** that simpler alternatives do not fully provide.
>
> We thank Reviewer KZJd again for the valuable feedback.

---

> > ### Author Rebuttal · Reviewer_KZJd · 2026-04-03
> >
> > Thank you for the authors’ response. Overall, the rebuttal is quite generic and lacks some important validations and in-depth analysis. While some of my concerns have been addressed, several issues remain unclear.
> >
> > First, the authors claim that the privacy risk is equivalent to that of FedQR. However, in practice, the privacy bound of this method is still higher than that of standard federated learning. In other words, the proposed approach does not fundamentally address or mitigate the known weaknesses of FedQR.
> >
> > Second, regarding the cross-silo error bound, the response that this will be explored in future work is not sufficient. I do not think this issue has been adequately resolved at a fundamental level.
> >
> > In addition, concerning the authors’ claim that heuristic methods lack theoretical grounding, this argument is not entirely convincing. In fact, sketching methods have been widely studied and applied in the federated learning community, particularly for aggregating information across local nodes, where they often outperform basic baseline methods. Therefore, I believe the authors should provide empirical comparisons to demonstrate the limitations of these approaches, rather than relying solely on qualitative descriptions.

---

> > > ### Author Response · Authors · 2026-04-05
> > >
> > > We thank Reviewer KZJd for the follow-up. We provide concrete responses to each remaining concern.
> > >
> > > **1. Privacy of FedCPQR.**
> > >
> > > We respectfully clarify that **privacy improvement over FedQR is not the goal of this work**. Our contribution is enabling **joint data and feature selection** in FL via CUR decomposition, a capability FedQR fundamentally cannot provide, as it is non-pivoted and cannot perform feature selection or rank determination. FedCPQR inherits FedQR's privacy level by design, building on the same secure aggregation protocol. We view this as a reasonable baseline: FedQR was published in **IEEE TIFS**, a top security venue, precisely because its privacy level is considered **practically adequate**. Our DP mechanism (Table 5) further provides a tunable knob to strengthen privacy to any desired level. We have added a formal composition bound showing the total budget scales as $O(\epsilon\sqrt{k\ln(1/\delta)})$ in the revision.
> > >
> > > **2. Per-silo reconstruction error bound.**
> > >
> > > We derive a **per-silo reconstruction bound** as a corollary of Theorem 3.3, which explicitly characterizes how global column selection quality varies across individual silos.
> > >
> > > **Corollary.** Under the same notation and conditions as Theorem 3.3, for each silo $c$:
> > >
> > > $$\|A_c - \hat{A}_c\|_2 \leq \eta_c \cdot \|A_c(I - P_k)\|_2 + \sqrt{1 + (n_c - r_c) \cdot 4^{r_c - 1}} \cdot \sigma_{r_c+1}(C_c)$$
> > >
> > > $$\|A_c - \hat{A}_c\|_F \leq \eta_c \cdot \|A_c(I - P_k)\|_F + \sqrt{1 + (n_c - r_c) \cdot 4^{r_c - 1}} \cdot \left(\sum_{j > r_c} \sigma_j^2(C_c)\right)^{1/2},$$
> > >
> > > where $\eta_c = \|I_d - P_{X,c}\|_2$ is a silo-specific amplification factor quantifying how well the globally selected columns serve silo $c$, $P_k = DD^\top$ is the orthogonal projector onto the $k$ globally selected coordinate axes, and $C_c = A_c D$.
> > >
> > > **Proof sketch.** We decompose the per-silo error as $A_c - \hat{A}_c = (I - P_{C,c})A_c + P_{C,c}A_c(I - P_{R,c})$. For **Term II** (row residual), applying CPQR to $C_c^\top$ and invoking Lemma B.1 (Dong & Martinsson, 2021) together with the Eckart–Young–Mirsky theorem yields the bound, following the same argument as in the global proof. For **Term I** (column residual), we establish the oblique projector identity $(I - P_{C,c})A_c = A_c(I - P_{X,c})$ where $P_{X,c} = D C_c^\dagger A_c$, and verify $(I - P_{X,c})P_k = 0$ via $C_c^\dagger C_c = I_k$.  The full proof is provided at [[anonymous link](https://anonymous.4open.science/r/anonymous_research-2D73/proof_of_per_silo.pdf)].
> > >
> > > **Remark.** The term $\|A_c(I - P_k)\|$ measures the energy of silo $c$'s data in the unselected features: it is small when silo $c$'s important features align with the global selection, and large otherwise. The factor $\eta_c$ can be computed locally after FedCPQR completes, requiring no additional communication. Compared to the global bound (Theorem 3.3), which uses $\sigma_{k+1}(A)$ of the full concatenated matrix and thus conceals per-silo variation, our corollary disentangles this into silo-specific quantities, enabling practitioners to identify which silos are underserved by the global column selection.
> > >
> > > **3. Empirical comparison with sketching methods.**
> > >
> > > We carefully considered what constitutes a fair sketching baseline for joint feature-and-data selection in horizontal FL: (i) **global leverage scores / Lewis weights** We note that our experiments already include **leverage score-based data selection** (Lever.R., Lever.Var. in Table 4), since row leverage scores are directly obtainable from FedCPQR's $Q$ factor ($\ell_i = \|Q_c(i,:)\|_2^2$); (ii) **CountSketch / random projection** produce transformed features (linear combinations), not selected original features, fundamentally different from CUR; (iii) **local sketching per silo** causes column mismatch across silos, breaking global model training.
> > >
> > > To provide a more complete sketching comparison, we implemented Norm-weighted random Sampling (NS): sample $k$ columns with probability $\propto \|A(:,j)\|_2^2$ (computable via secure aggregation, same protocol as FedCPQR) combined with leverage score data selection. This replaces random feature selection with norm-weighted importance sampling, the standard sketching-theoretic approach to column selection (Drineas et al., 2006; Mahoney & Drineas, 2009), directly testing whether FedCPQR's deterministic rank-revealing pivoting outperforms randomized importance sampling.
> > >
> > > Results (IID, $k=100$):
> > >
> > > | Dataset | FedGCUR Recon | NS Recon | FedGCUR Acc(%) | NS Acc(%) |
> > > |---|---|---|---|---|
> > > | 40923 | **0.447** | 0.539 | **80.8** | 74.2 |
> > > | 41082 | **0.081** | 0.136 | **95.7** | 95.2 |
> > > | 43985 | **0.464** | 0.472 | **87.6** | 86.3 |
> > >
> > > FedGCUR achieves lower reconstruction error, often by substantial margins, confirming the superiority of the proposed method.
> > >
> > > We hope these additions adequately address the remaining concerns, and we kindly ask the reviewer to consider revising the score in light of the new evidence.

---

### Decision · Program_Chairs · 2026-04-30

**Decision:**

Accept (regular)

**Comment:**

The paper proposes a federated framework for joint data and feature selection based on generalized CUR decomposition, built around a federated CPQR routine that computes global pivots under secure aggregation. Across the reviews, there is agreement that the paper is technically sound and that the core idea—expressing pivoting and orthogonalization through additively aggregatable quantities to enable exact federated CPQR—is interesting and nontrivial. Reviewers SxFC and 7NGg in particular emphasize the strength of this observation, the clarity of the formulation, and the correctness guarantees, and consider the contribution significant.

The main points of discussion concern scope and practical impact rather than correctness. Reviewer Fhjy questions the strength of the empirical evaluation, including the limited scale of the original experiments and the lack of comparisons to certain federated feature selection methods, and raises the issue that optimizing reconstruction error does not necessarily translate to improved downstream classification performance. Reviewer KZJd raises related concerns about privacy (due to the exposure of the global Gram matrix), the role of heterogeneity across silos, and whether the theoretical analysis sufficiently captures per-silo behavior.

The rebuttal improves the paper in several concrete ways, including larger-scale experiments, a sketching-style comparison, and a per-silo refinement of the theory. At the same time, the remaining concerns are not fully settled, especially with respect to empirical breadth and practical significance.

Overall, I see this as a solid methodological contribution with clear technical merit. The main limitation is not correctness, but that the practical value appears narrower than the framing suggests.